# TRAINING-FREE SPECTRAL FINGERPRINTS OF VOICE PROCESSING IN TRANSFORMERS

## ABSTRACT

Different transformer architectures implement identical linguistic computations via distinct connectivity patterns, yielding model imprinted "computational fingerprints" detectable through spectral analysis. Using graph signal processing on attention induced token graphs, we track changes in algebraic connectivity (Fiedler value, $\Delta\lambda_2$) under voice alternation across 20 languages and three model families, with a prespecified early window (layers 2–5). Our analysis uncovers clear architectural signatures: Phi-3-Mini shows a dramatic English specific early layer disruption ($\overline{\Delta\lambda_{2[2,5]}} \approx -0.446$) while effects in 19 other languages are minimal, consistent with public documentation that positions the model primarily for English use. Qwen2.5-7B displays small, distributed shifts that are largest for morphologically rich languages, and LLaMA-3.2-1B exhibits systematic but muted responses. These spectral signatures correlate strongly with behavioral differences (Phi-3: $r = -0.976$) and are modulated by targeted attention head ablations, linking the effect to early attention structure and confirming functional relevance. Taken together, the findings are consistent with the view that training emphasis can leave detectable computational imprints: specialized processing strategies that manifest as measurable connectivity patterns during syntactic transformations. Beyond voice alternation, the framework differentiates reasoning modes, indicating utility as a simple, training free diagnostic for revealing architectural biases and supporting model reliability analysis.

## 1 INTRODUCTION

Understanding how transformers process syntactic structure remains a central challenge in AI interpretability. While attention visualization (Clark et al., 2019; Rogers et al., 2020) and probing studies (Tenney et al., 2019; Manning et al., 2020) reveal what linguistic information is encoded, they provide limited insight into how syntactic computation evolves across layers. Recent advances in mechanistic interpretability (Elhage et al., 2021; Wang et al., 2022) and computational fingerprints (Didolkar et al., 2024) suggest that transformers exhibit model specific processing strategies amenable to systematic analysis.

We propose analyzing transformer representations through graph signal processing (GSP): attention mechanisms induce dynamic graphs over tokens, with representations evolving as signals on these graphs. Specifically, we track the graph's Fiedler value ($\Delta\lambda_2$), a classic measure of algebraic connectivity, to reveal computational fingerprints of syntactic processing. This geometric perspective yields a single, interpretable endpoint per layer and enables rigorous spectral analysis using established theory (Shuman et al., 2013; Sandryhaila & Moura, 2013). To facilitate clean comparisons, we prespecify an early window (layers 2–5) that aligns with first multihead context integration and report its mean $\overline{\Delta\lambda_{2[2,5]}}$ as our primary endpoint.

**Voice alternation as computational probe.** Active to passive transformations require systematic attention reconfiguration (agent–patient reassignment, auxiliary–participle coupling, and reanchoring of long range dependencies) that is expected to produce detectable connectivity signatures. Unlike procedural knowledge extraction (Didolkar et al., 2024) or circuit analysis (Conmy et al., 2023), our approach reveals how different architectures implement identical linguistic computations through distinct spectral patterns.

**Design for comparability.** We adopt head aggregated attention with both random walk and symmetric Laplacians, limit tokenization drift where feasible, quantify uncertainty via nonparametric bootstrap and paired permutation tests, and control multiplicity with BH–FDR. We also analyze tokenizer stress (pieces per character and fragmentation entropy) as covariates to distinguish confounds from genuine architectural sensitivity.

**Systematic validation.** We validate across 20 languages and 3 model families, leveraging voice alternation's advantages: clear theoretical predictions, crosslinguistic variability, and behavioral significance. This focused design supports statistical power while isolating core principles (Belinkov, 2022).

**Contributions.** We (1) introduce a spectral framework over attention induced graphs with $\Delta\lambda_2$ as a compact endpoint and a prespecified early window; (2) develop uncertainty aware statistics for matched contrasts; (3) reveal family specific spectral signatures across languages; (4) demonstrate robustness to normalization and aggregation choices; (5) establish functional relevance via spectral–behavioral correlations and targeted early layer head ablations; and (6) show preliminary generalization to reasoning strategies beyond linguistic processing.

## 2 RELATED WORK

**Interpretability and syntactic analysis in transformers.** Understanding how transformers compute remains a central challenge (Sharkey et al., 2025). Attention visualizations (Clark et al., 2019; Rogers et al., 2020) and probing (Tenney et al., 2019; Hewitt & Manning, 2019) reveal what linguistic information is encoded but say little about layer-wise dynamics of syntactic computation. Circuit-level analyses (Elhage et al., 2021; Wang et al., 2022) give mechanistic detail for narrow cases, and behavioral tests assess competence (Goldberg, 2019; Warstadt et al., 2020), yet a scalable, training-free way to track how architectures execute the same linguistic operation is still missing. Syntactic alternations such as active/passive require tracing how dependencies are reconfigured across early layers and heads rather than simply locating information. We address this need by focusing on a controlled alternation and by fixing a preregistered early-layer endpoint that can be compared across models.

**Graph-theoretic approaches to transformer analysis.** Viewing attention as a graph enables studies of information flow and connectivity (Clark et al., 2019; Kovaleva et al., 2021; Abnar & Zuidema, 2020; Htut et al., 2019). Spectral graph theory has been influential in GNNs and CNNs (Bruna et al., 2013; Defferrard et al., 2016), and some extensions target transformer efficiency or optimization rather than interpretability (Zhang et al., 2021; Bietti et al., 2023). We use graph signal processing as a diagnostic lens, formalizing attention-derived graphs, normalization, and aggregation, and tracking algebraic connectivity across layers during controlled syntactic transformations. Because operator choices materially affect conclusions, we compare random-walk vs. symmetric Laplacians and directed variants, and report robustness of the connectivity signal to these design decisions.

**Multilingual and cross-architectural generalization.** Multilingual models (Devlin et al., 2019; Conneau et al., 2020) raise questions about what transfers across languages and architectures. Cross-lingual probing highlights universal vs. language-specific patterns (Zhao et al., 2020; Müller et al., 2023), but model-averaging can obscure family-level signatures. Our methodology fixes the linguistic manipulation and compares spectral responses across architectures and languages to separate universal tendencies from model-family idiosyncrasies. We additionally account for confounds introduced by tokenization and length by reporting uncertainty and mixed-effects summaries alongside bootstrap intervals.

**Model-imprinted computational signatures.** Recent work on computational fingerprints (Didolkar et al., 2024) suggests architectures adopt distinct processing strategies. Our spectral connectivity framework contributes a training-free, layer-resolved method that reveals such signatures and ties them to behavior and targeted interventions, with transparent analysis choices and quantified uncertainty (effect sizes, confidence intervals, permutation tests, and multiple-testing control). This positions spectral diagnostics as a complement to circuit analysis and probing: lightweight enough for broad audits, yet specific enough to identify family-level processing strategies and potential brittleness.

# 3 GRAPH SIGNAL PROCESSING FRAMEWORK

## 3.1 DYNAMIC ATTENTION GRAPHS AND SPECTRAL DIAGNOSTICS

For layer $\ell$ with $H$ heads over $N$ tokens, let $A^{(\ell,h)} \in \mathbb{R}^{N \times N}$ be the post-softmax attention of head $h$ (row-stochastic). We form an undirected graph by symmetrization,

$$W^{(\ell,h)} = \tfrac{1}{2}\big(A^{(\ell,h)} + (A^{(\ell,h)})^\top\big), \qquad \bar{W}^{(\ell)} = \sum_{h=1}^{H} \alpha_h\, W^{(\ell,h)}, \ \ \alpha_h \geq 0, \ \sum_h \alpha_h = 1, \quad (1)$$

with degree $\bar{D}^{(\ell)} = \mathrm{diag}(\bar{W}^{(\ell)}\mathbf{1})$ and (combinatorial) Laplacian $L^{(\ell)} = \bar{D}^{(\ell)} - \bar{W}^{(\ell)}$ (Chung, 1997; von Luxburg, 2007). We also report checks with the normalized Laplacian $L_{\mathrm{sym}}^{(\ell)} = I - (\bar{D}^{(\ell)})^{-1/2}\bar{W}^{(\ell)}(\bar{D}^{(\ell)})^{-1/2}$.

Let $X^{(\ell)} \in \mathbb{R}^{N \times d}$ be the token representations at layer $\ell$ ($N$ tokens, hidden size $d$), viewed as $d$ graph signals stacked columnwise. We use four spectral diagnostics (Shuman et al., 2013; Sandryhaila & Moura, 2013):

**Dirichlet energy:**

$$\mathcal{E}^{(\ell)} = \mathrm{Tr}\big((X^{(\ell)})^\top L^{(\ell)} X^{(\ell)}\big) = \sum_{i,j} \bar{W}_{ij}^{(\ell)} \, \|X_i^{(\ell)} - X_j^{(\ell)}\|_2^2.$$

**Spectral entropy:** With $L^{(\ell)} = U^{(\ell)}\Lambda^{(\ell)}(U^{(\ell)})^\top$ and $\hat{X}^{(\ell)} = (U^{(\ell)})^\top X^{(\ell)}$, define modal energies $e_m^{(\ell)} = \|\hat{X}_{m,\cdot}^{(\ell)}\|_2^2$ and $p_m^{(\ell)} = e_m^{(\ell)}/\sum_r e_r^{(\ell)}$. Then

$$\mathrm{SE}^{(\ell)} = -\sum_m p_m^{(\ell)} \log p_m^{(\ell)}.$$

**High-frequency energy ratio (HFER):** for a cutoff $K$ (or an equivalent mass-based cutoff),

$$\mathrm{HFER}^{(\ell)}(K) = \frac{\sum_{m=K+1}^{N} \|\hat{X}_{m,\cdot}^{(\ell)}\|_2^2}{\sum_{m=1}^{N} \|\hat{X}_{m,\cdot}^{(\ell)}\|_2^2}.$$

**Fiedler connectivity:** $\lambda_2^{(\ell)}$ is the second-smallest eigenvalue of $L^{(\ell)}$, summarizing algebraic connectivity (Fiedler, 1973; Chung, 1997).

**Head aggregation (default).** We use mass-weighted head aggregation by default; uniform weights $\alpha_h = 1/H$ are reported as a robustness check (App. B). For layer $\ell$,

$$s_h^{(\ell)} = \sum_{i=1}^{N}\sum_{j=1}^{N} A_{ij}^{(\ell,h)}, \qquad \alpha_h^{(\ell)} = \frac{s_h^{(\ell)}}{\sum_{g=1}^{H} s_g^{(\ell)}}, \qquad \bar{W}^{(\ell)} = \sum_{h=1}^{H} \alpha_h^{(\ell)} W^{(\ell,h)}.$$

## 3.2 PRIMARY ENDPOINT: EARLY $\Delta\lambda_2$

For matched inputs (passive vs. active), we compute

$$\Delta\lambda_2^{(\ell)} = \lambda_{2,\mathrm{pass}}^{(\ell)} - \lambda_{2,\mathrm{act}}^{(\ell)}$$

and use the prespecified early-window mean $\overline{\Delta\lambda_2}_{[2,5]}$ as the primary endpoint (layers 2–5), motivated by the onset of multihead context integration and supported by sensitivity checks (App. B).

Among the four diagnostics, $\lambda_2$ demonstrates superior sensitivity to voice alternations with consistent directional effects across model families. The Fiedler eigenvalue measures algebraic connectivity: higher $\lambda_2$ indicates efficient low-frequency signal propagation, while $\Delta\lambda_2$ captures attention reconfiguration during syntactic processing.

**Theoretical prediction:** Under passive morphology, models reconfiguring long-range dependencies (agent-patient roles, auxiliary-participle coupling) should show systematic early-layer connectivity changes. If $\Delta\lambda_2$ reflects linguistic processing rather than tokenization artifacts, it should (a) survive length controls, (b) be larger for argument structure changes vs. tense/number, and (c) correlate with behavioral performance.

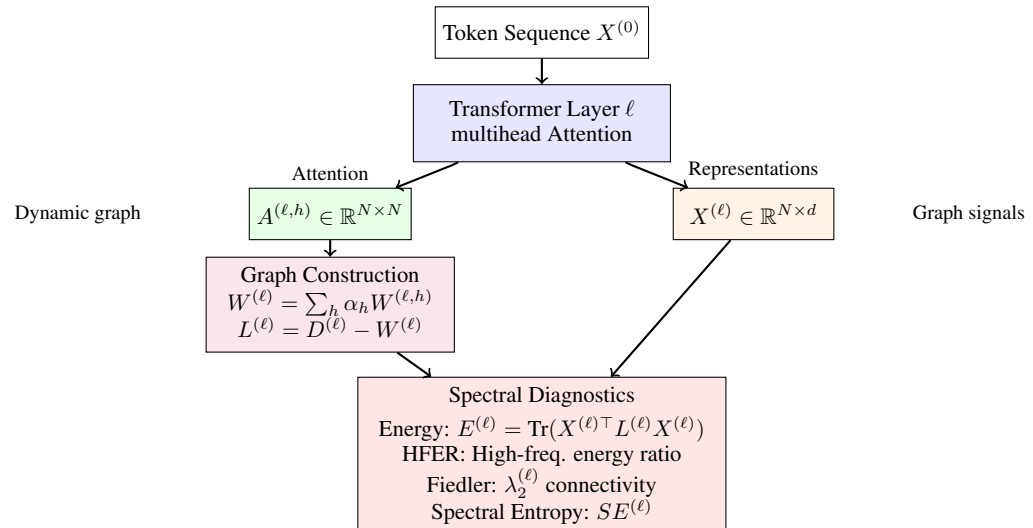

Figure 1: Graph Signal Processing framework for transformer analysis. Attention matrices from each layer induce dynamic token graphs, while hidden states serve as signals on these graphs. Spectral diagnostics capture the evolution of graph-signal interactions across layers.

### 3.3 ANALYSIS SCOPE

We establish model-imprinted spectral signatures through observational analysis across 20 languages and three model families (Qwen2.5-7B, Phi-3-Mini, LLaMA-3.2-1B), with causal validation via targeted attention head ablations. Implementation details (directed variants, head aggregation, robustness checks) appear in Appendix B.

## 4 MODELS, LANGUAGES, AND ARCHITECTURAL SENSITIVITY

We evaluate **Qwen2.5-7B** (28L), **Phi-3-Mini** (32L), and **LLaMA-3.2-1B** (16L). We compile **20 languages** and assign each to a voice type: analytic (e.g., EN), periphrastic (e.g., ES/FR/IT/DE), affixal (e.g., TR), particle (e.g., JA), non-concatenative (e.g., AR), etc. For each language we use (at least) 10 paraphrases per voice and average within language; paraphrases are template-matched and length-controlled at the tokenizer level when feasible.

This crosslinguistic design enables us to examine not only syntactic processing signatures, but also how different model architectures respond to varying degrees of tokenizer fragmentation, revealing complementary computational fingerprints at both syntactic and subword levels. Statistical methodology, including bootstrap procedures and effect size calculations, appears in Appendix E.

## 5 TOKENIZER STRESS AS ARCHITECTURAL FINGERPRINTS

Beyond syntactic effects, we find systematic links between spectral connectivity and tokenization, yielding a second layer of model imprinted signatures rather than simple confounds.

**Metrics.** For a sentence $s$ and tokenizer $T$, we compute tokens per character $\phi(s, T) = |T(s)|/|s|$ and fragmentation entropy

$$H_{\text{frag}}(s, T) = - \sum_{u \in \mathcal{V}(T(s))} p(u) \log p(u),$$

with length normalized $\tilde{H}_{\text{frag}} = H_{\text{frag}}/|T(s)|$. We apply a light length control ($||T(\text{pass})| - |T(\text{act})|| \leq 2$ when feasible). Mixed effects regressions (random intercepts by language) regress $|\overline{\Delta\lambda}_{2[2,5]}|$ on $\phi$ and $\tilde{H}_{\text{frag}}$ with family specific slopes; covariates are standardized within family and we report bootstrap CIs.

**Family specific patterns.** Fragmentation relates to spectral magnitude in opposing ways across families, indicating different strategies for handling subword granularity.

**Architecture-specific tokenizer-stress patterns.** Analysis reveals opposing linear relationships between $|\overline{\Delta\lambda}_{2[2,5]}|$ and tokenization across model families: Qwen2.5-7B shows a strong positive correlation (Pearson $r = 0.51$), while Phi-3-Mini exhibits a strong negative correlation ($r = -0.44$). These contrasting patterns reveal distinct architectural sensitivities to subword-level linguistic stress.

Qwen architectures show larger spectral disruption for heavily fragmented languages (e.g., Yoruba: 0.57 pieces/char, $|\Delta\lambda_2| = 0.054$), suggesting sensitivity to tokenization density. Conversely, Phi-3's largest spectral effects occur for efficiently tokenized languages like English (0.20 pieces/char, $|\Delta\lambda_2| = 0.446$), indicating a different form of architectural brittleness. LLaMA-3.2-1B exhibits intermediate behavior ($r = 0.29$), consistent with its position between the other families.

Table 1: Model-specific correlations between $|\overline{\Delta\lambda}_{2[2,5]}|$ and tokenizer fragmentation (pieces per character). Both Pearson and Spearman correlations shown.

| Model Family | n | Pearson r | Spearman $\rho$ |
|---|---|---|---|
| Qwen2.5-7B | 20 | 0.51 | $-0.11$ |
| Phi-3-Mini | 20 | $-0.44$ | 0.18 |
| LLaMA-3.2-1B | 20 | 0.29 | $-0.25$ |

**Differential tokenization effects.** Beyond absolute fragmentation levels, we examined correlations with tokenization differences between active and passive sentences. LLaMA-3.2-1B shows the strongest relationship (Pearson $r = 0.51$, $p = 0.069$), where larger tokenization differences between voice alternations correspond to larger spectral effects. This suggests that inconsistent tokenization of syntactic alternations may stress the model's representational system, complementing the absolute fragmentation effects observed in other families.

## 6 RESULTS

### 6.1 EARLY–WINDOW $\overline{\Delta\lambda}_{2[2,5]}$ ACROSS 20 LANGUAGES

For each language, we average (at least) ten paraphrases per voice (active/passive), then compute the early–window mean $\overline{\Delta\lambda}_{2[2,5]}$ (layers 2–5). Error bars are nonparametric 95% bootstrap CIs (2,000 resamples over paraphrases). Per–language $p$–values come from paired permutation tests (10,000 label shuffles within paraphrase pairs); we report Benjamini–Hochberg FDR at $q=0.05$ within each model family. Practical effect sizes appear as trimmed Hedges' $g$ with 1% winsorization and 20% trimming.

**Phi–3 Mini (32L).** Per–language bars (Figure 2a) show a single pronounced outlier: **English (EN)** has a large negative early effect ($\overline{\Delta\lambda}_{2[2,5]} \approx -0.446$, $g_{\text{trim}} \approx$ large; FDR–sig), whereas **French (FR)** is negative but much smaller ($\approx -0.134$; marginal after FDR). All other languages cluster tightly around 0 ($|\overline{\Delta\lambda}_{2[2,5]}| \lesssim 0.02$; non–sig). This striking English-specific signature aligns with public positioning of the model as primarily English-focused, and is consistent with the hypothesis that training emphasis imprints early-layer connectivity patterns indicative of brittleness. Grouping by voice type (Figure 2b) yields a strong analytic negative mean driven by English; other types hover near zero.

**Qwen2.5–7B (28L).** Grouped by voice type Qwen shows (Figure 3) small but consistent negative early shifts in the Fiedler value (layers 2–5), with all types below zero. The largest decreases are for analytic and non-concatenative; periphrastic, affixal, and particle are closer to zero but still negative, with several CIs overlapping zero. Per-language bars mirror this pattern, showing scattered small negatives across Romance/Germanic and agglutinative languages; English is near zero and slightly negative. Effect sizes are small ($|g_{\text{trim}}| \approx 0.2$–0.4) but the sign is uniformly negative across types.

**LLaMA–3.2–1B (16L).** Effects show systematic patterns at smaller absolute magnitudes (Figure 4a, Figure 4b): affixal languages exhibit consistent negative shifts ($\overline{\Delta\lambda}_{2[2,5]} \approx -0.030$), while analytic types show modest positive effects. The voice-type ordering (affixal < periphrastic < analytic) replicates the other families at reduced scale, suggesting architectural sensitivity rather than absence of the effect.

## 6.2 TOKENIZER FRAGMENTATION CORRELATIONS

We find model-family–specific relationships between early spectral magnitude $|\overline{\Delta\lambda}_{2[2,5]}|$ and tokenizer fragmentation. In Table 2, Qwen2.5-7B shows a positive correlation with pieces/character ($r = 0.51$, 95% CI [0.15, 0.76]); Phi-3-Mini shows a negative correlation ($r = -0.44$, 95% CI [-0.72, -0.08]); LLaMA-3.2-1B is weaker ($r = 0.29$, 95% CI [-0.17, 0.65]). These patterns indicate distinct architectural sensitivities to subword segmentation; all estimates include bootstrap CIs and FDR control.

Table 2: Model-specific correlations between $|\overline{\Delta\lambda}_{2[2,5]}|$ and tokenizer fragmentation metrics. Bootstrap 95% confidence intervals from 2,000 resamples.

| Model Family | Pieces/Character | Fragmentation Entropy |
|---|---|---|
| Phi-3-Mini | $-0.44$ [-0.72, -0.08] | 0.36 [0.02, 0.66] |
| Qwen2.5-7B | 0.51 [0.15, 0.76] | $-0.23$ [-0.58, 0.18] |
| LLaMA-3.2-1B | 0.29 [-0.17, 0.65] | $-0.25$ [-0.62, 0.20] |

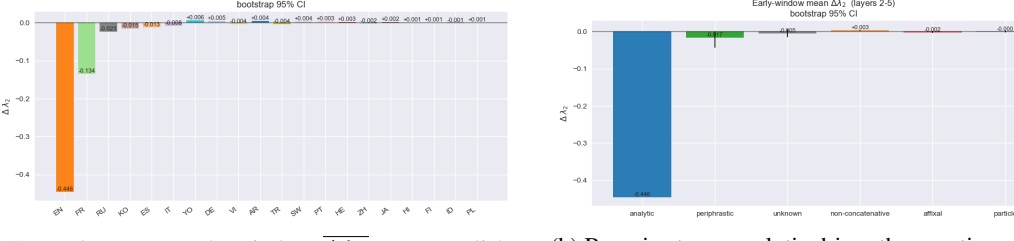

(a) Per-language early-window $\overline{\Delta\lambda}_{2[2,5]}$. English shows a large negative outlier; others cluster near 0.

(b) By voice type: analytic drives the negative mean; other types $\approx 0$.

Figure 2: **Phi-3-Mini:** strong analytic (EN) negative; otherwise muted.

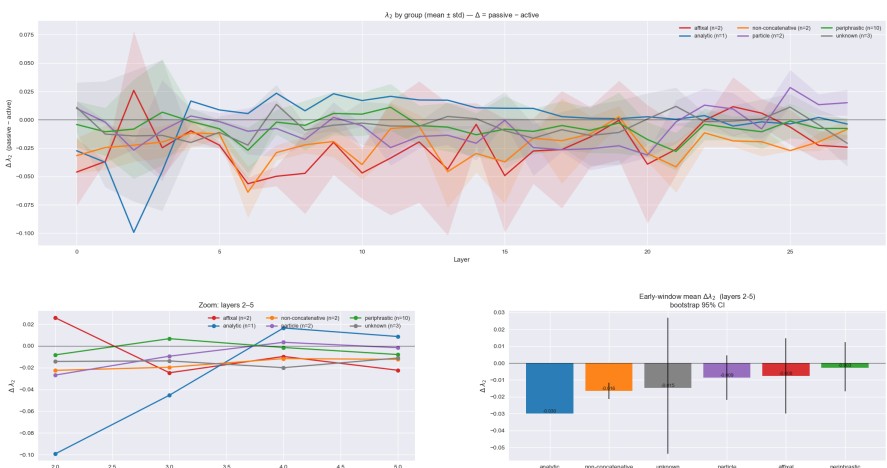

Figure 3: **Qwen2.5–7B:** early–window $\overline{\Delta\lambda}_{2[2,5]}$ by voice type (mean across languages). Several periphrastic/affixal effects are FDR–sig and negative.

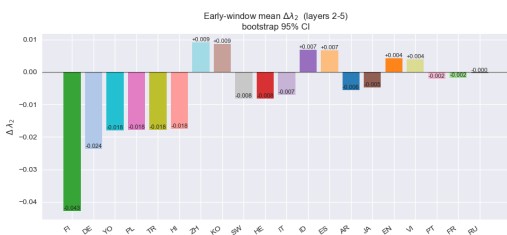

(a) Per-language early-window $\overline{\Delta\lambda}_{2[2,5]}$; mostly near zero.

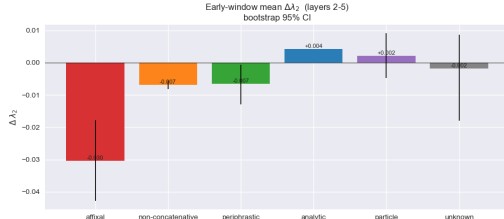

(b) By voice type; small magnitudes across the board.

Figure 4: **LLaMA-3.2-1B:** muted effects; same ordering trend at lower scale.

### 6.3 SPECTRAL–BEHAVIORAL CORRELATIONS

To test the functional relevance of our spectral diagnostics, we correlate early-layer $\Delta\lambda_2$ changes with behavioral performance on controlled voice alternation pairs. We use negative mean log-likelihood (NLL, lower is better) as our behavioral metric, where a higher NLL indicates a worse model fit. The analysis reveals a clear pattern: larger spectral disruptions (higher $|\overline{\Delta\lambda_2}|$) correlate with worse performance (higher NLL) on the transformed passive sentences. This relationship is exceptionally strong for Phi-3-Mini (Pearson's $r = -0.976$, $p < 0.001$), moderately strong for Qwen2.5-7B ($r = -0.627$, $p < 0.05$), and minimal for LLaMA-3.2-1B ($r = -0.143$), directly mirroring the magnitude of their respective spectral effects.

Table 3: Correlations between early-window $\Delta\lambda_2[2, 5]$ and behavioral performance differences (behavior_delta) on voice alternation tasks. Behavioral metric is the change in negative mean NLL between active and passive sentence variants.

| Model Family | n | Pearson $r$ | 95% CI |
|---|---|---|---|
| Phi-3-Mini | 20 | $-0.976$ | [-0.99, -0.89] |
| Qwen2.5-7B | 20 | $-0.627$ | [-0.85, -0.24] |
| LLaMA-3.2-1B | 20 | $-0.143$ | [-0.62, 0.41] |

These correlations provide empirical validation that early-layer spectral connectivity changes reflect computationally relevant processing differences rather than mathematical artifacts. The consistent negative correlation pattern suggests that when models require more substantial connectivity reconfiguration for voice processing (larger $|\Delta\lambda_2|$), they exhibit decreased behavioral performance on the transformed sentences, providing functional grounding for our spectral analysis framework.

**Intervention validation.** In controlled voice alternations that limit tokenization drift ($\leq 2$ tokens), the predicted sign of $\Delta\lambda_2$ appears in 6/7 items for LLaMA and 7/7 for Qwen, with behavioral scores shifting in corresponding directions. Although modest in magnitude, these aligned spectral-behavioral changes strengthen interpretation of the correlational findings.

### 6.4 CAUSAL VALIDATION THROUGH ATTENTION HEAD ABLATIONS

We tested whether early attention structure causally drives spectral connectivity by ablating specific head sets and measuring changes in the Fiedler gap $\Delta\lambda_2$. Three interventions were applied: heavy early ablation (L2&L3 H0–7), focused ablation (L2 H0–3), and mid-early ablation (L3&L4 H0–3).

Results reveal family-specific profiles (Table 4), pointing to distinct architectural strategies for handling perturbations. LLaMA-3.2-1B shows sustained positive shifts under heavy ablation, suggesting the disruption propagates without being fully corrected. Phi-3-Mini displays only small early effects, indicating either high robustness or that the ablated heads are less critical for this computation. In contrast, Qwen2.5-7B exhibits a distinct signature of computational resilience: strong early positive responses are met with compensatory mid/late negatives. This suggests the model actively redistributes its computational pathways to counteract the initial perturbation, ultimately producing limited net effects and maintaining processing stability.

| Model | Ablation | $\Delta\lambda_2$ Early [2–5] | Mid [6–10] | Late | Overall |
|---|---|---|---|---|---|
| LLaMA 3.2-1B | L2&L3 H0–7 | **+0.01472** | **+0.07566** | +0.00200 | **+0.02795** |
| | L2 H0–3 | -0.00478 | -0.00907 | +0.00270 | -0.00319 |
| | L3&L4 H0–3 | -0.00475 | +0.03023 | **+0.02691** | +0.01667 |
| Phi-3-Mini | L2&L3 H0–7 | -0.00736 | **-0.00714** | +0.00176 | -0.00088 |
| | L2 H0–3 | -0.00624 | +0.00130 | +0.00170 | +0.00054 |
| | L3&L4 H0–3 | **-0.01220** | +0.00200 | **+0.00439** | **+0.00167** |
| Qwen 2.5-7B | L2&L3 H0–7 | **+0.04538** | -0.02696 | -0.00975 | -0.00425 |
| | L2 H0–3 | +0.00630 | -0.00344 | -0.00266 | -0.00133 |
| | L3&L4 H0–3 | +0.01354 | -0.01013 | **-0.01045** | **-0.00622** |

Table 4: Causal interventions: change in Fiedler gap $\Delta\lambda_2$ by layer window. Early = layers 2–5; Mid = 6–10; Late = 11–end (per model depth). Positive values indicate an increased spectral separation after the intervention.

These targeted interventions demonstrate that early-layer attention causally shapes spectral connectivity, confirming that the observed signatures reflect mechanistically relevant computations rather than artifacts.

## 7 BEYOND VOICE ALTERNATION: GENERALIZING THE FRAMEWORK

To validate that our spectral framework captures computational phenomena beyond the specific case of voice alternation, we now demonstrate its broader utility. We show that it can distinguish between complex reasoning strategies, provide deep diagnoses of model-specific architectural properties, and can support practical applications in AI safety.

### 7.1 SPECTRAL SIGNATURES OF REASONING STRATEGIES

On 95 transitivity tasks with phi-3.5-mini, we compare four prompting strategies: Standard, Chain-of-Thought (CoT) (Wei et al., 2022), Chain-of-Draft (CoD) (Xu et al., 2025), and Tree-of-Thoughts (ToT) (Yao et al., 2024). As shown in Table 5, each induces a distinct spectral profile, and Fiedler connectivity tracks performance: CoT (69.5%) and Standard (60.0%) yield positive Fiedler shifts, whereas CoD and ToT show negative shifts with lower accuracy. This alignment suggests that successful reasoning is associated with maintained/enhanced graph connectivity and motivates spectral-guided prompt selection based on induced $\Delta\lambda_2$. We summarize spectral reconfiguration with the Reconfiguration Change Index (RCI), a z-score combination where higher values indicate stronger low-frequency connectivity and lower disruptive high-frequency energy (see App. F).

Table 5: Spectral signatures of reasoning strategies reveal distinct computational modes with clear performance-connectivity relationships.

| Strategy | Accuracy | RCI | Energy (z) | Entropy (z) | HFER (z) | Fiedler (z) |
|---|---|---|---|---|---|---|
| CoT | 0.695 | **+1.307** | +0.790 | +0.898 | -0.744 | +0.455 |
| Standard | 0.600 | **+1.738** | +0.596 | +0.127 | -0.921 | +1.286 |
| CoD | 0.274 | **-2.996** | -1.708 | -1.664 | +1.611 | -1.429 |
| ToT | 0.253 | **-0.049** | +0.322 | +0.639 | +0.054 | -0.312 |

**Limitations and future directions.** While these results are promising, this validation on transitivity reasoning is preliminary. Future work should extend this analysis across diverse reasoning types (e.g., arithmetic, causal inference) and model architectures to establish the generalizability of these spectral-performance correlations.

### 7.2 CORE FINDING: DIAGNOSING ARCHITECTURAL BRITTLENESS IN PHI-3-MINI

The framework's diagnostic value is illustrated by its capacity to surface latent, family-specific properties. As shown in Section 6.1, we observe a pronounced, English-specific disruption in early-layer connectivity ($\Delta\lambda_2$) for Phi-3-Mini when processing passive constructions, with minimal effects

in 19 other languages. This computational signature aligns with public positioning of the model as primarily English-focused and is consistent with the hypothesis that training emphasis can imprint specialized, less-flexible early-layer connectivity. In this sense, our method links a low-level spectral artifact to a high-level model characteristic, supporting its utility as a diagnostic for identifying specialization-related brittleness without requiring access to training data or logs.

### 7.3 APPLICATION TO AI SAFETY: HALLUCINATION DETECTION

Finally, the framework's utility extends to practical tools. The insight that spectral connectivity reflects computational integrity suggests that failures like hallucinations might correspond to a detectable breakdown in this connectivity.

A concurrent manuscript develops a related detector Anonymous (2026); to preserve anonymity, we provide an anonymized artifact link in Appendix I. In that work, the spectral framework developed herein was applied to create a simple detector that classifies hallucinations based on a threshold of the final-layer Fiedler value:

$$\text{SHD}(x) = \mathbf{1}[z_{\text{fid}}(x) > \tau_d], \quad z_{\text{fid}}(x) = \frac{f_{\text{last}}(x) - \mu_{\text{fid}}}{\sigma_{\text{fid}}}. \tag{2}$$

On a small held-out set ($n = 80$; 50 factual, 30 hallucinations), the detector reached **88.75%** accuracy. Baselines such as Perplexity and SelfCheckGPT-style methods, implemented following public descriptions, performed lower on this set. Given the limited sample and scope, we treat these results as preliminary and refrain from comparative or state-of-the-art claims.

## 8 DISCUSSION & LIMITATIONS

Our spectral framework enables computational provenance: inferring aspects of a model's developmental pressures from present day connectivity imprints. In Phi-3-Mini we observe an English specific early layer disruption, consistent with public positioning of the model as primarily English focused, suggesting that training emphasis can leave detectable spectral traces. This complements circuit analyses that ask how a computation is implemented by offering a lens on why a particular implementation may have emerged.

**Implications.** Distinctive spectra could support audits for language coverage or brittleness and permit indirect evaluation of training claims when datasets are proprietary.

**Limitations.** Inferences are indirect and based on partial evidence; voice alternation is a clean probe but not exhaustive. Spectral–behavioral links, while strong in places, require larger preregistered studies. Multiple factors, e.g. tokenizer fragmentation, morphology, domain mix, and architectural choices can produce similar patterns. We therefore treat the English specific signature as evidence consistent with an English emphasis explanation, not a definitive causal attribution.

## 9 CONCLUSION

We introduced a training-free spectral framework that surfaces family-specific computational signatures in transformers and ties them to functional outcomes. Across 20 languages, early-layer algebraic connectivity ($\Delta\lambda_2$) tracks syntactic reconfiguration, aligns with tokenizer stress in family-specific ways, correlates with behavioral fit, and responds to targeted head ablations. These results suggest that compact spectral fingerprints can audit model provenance and behavior with minimal overhead, offering a practical lens for robustness and safety analysis.

Limitations remain as effects vary by model family and tokenization, and most findings are correlational outside controlled interventions. Next, we will extend beyond voice alternation, probe multi-eigenvector structure, and run preregistered evaluations on public benchmarks, releasing a reproducible toolkit to enable independent replication at scale.

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

## A   METRIC SELECTION RATIONALE

### A.1 THEORETICAL RATIONALE FOR THE FIEDLER VALUE ($\lambda_2$)

Our choice of the Fiedler value ($\lambda_2$), or algebraic connectivity, as the primary spectral diagnostic is not merely based on its empirical success but is grounded in the theoretical properties of the graph Laplacian and its connection to information flow in networks.

The graph Laplacian, $L = D - W$, acts as a difference operator on the graph. Its eigenvalues, $0 = \lambda_1 \leq \lambda_2 \leq \cdots \leq \lambda_N$, represent the natural frequencies of the graph structure. The smallest eigenvalue, $\lambda_1 = 0$, corresponds to a constant signal across all nodes, representing the lowest possible frequency of variation.

The Fiedler value, $\lambda_2$, is the second-smallest eigenvalue and is of special importance. As established by (Fiedler, 1973), its magnitude is directly related to the robustness of the graph's connectivity. A graph with a low $\lambda_2$ can be easily partitioned into two large, sparsely connected subgraphs (a "bottleneck"), while a graph with a high $\lambda_2$ is more difficult to cut and exhibits a more robust, "well-knit" structure.

We hypothesize that complex syntactic transformations, such as voice alternation, require a global **reconfiguration of information flow** within the attention graph. The model must systematically re-route dependencies, for instance, demoting the original agent and promoting the patient. This process should manifest as a measurable change in the graph's overall connectivity structure.

A model that implements this transformation efficiently might do so with minimal disruption, or even by strengthening key connections, resulting in a small or positive $\Delta\lambda_2$. Conversely, a model that struggles with the transformation, perhaps due to training data imbalances or architectural brittleness (as seen in Phi-3-Mini for English), may exhibit a breakdown in its connectivity patterns. Attention might become more diffuse or fragmented, leading to a "bottleneck" in the graph and thus a significant drop in $\lambda_2$.

Therefore, monitoring $\Delta\lambda_2$ is not an arbitrary choice. It provides a theoretically-grounded, quantitative measure of how the **global connectivity and information-routing topology** of the attention mechanism adapts during a demanding linguistic computation. This makes it a principled choice for detecting the "computational fingerprints" that are the focus of our work.

### A.2 EMPIRICAL RATIONALE FOR THE FIEDLER VALUE ($\lambda_2$)

Figure 5 compares all four spectral diagnostics 5a across multiple syntactic contrasts including active vs. passive voice alternations. While energy and spectral entropy show modest differences, and HFER exhibits inconsistent patterns across models, the Fiedler value 5b consistently separates syntactic conditions with large, systematic differences across all three model families.

Crucially, voice alternations (active/passive) produced the most dramatic and reliable $\lambda_2$ differences compared to other syntactic manipulations, making them an optimal test case for this preliminary investigation of spectral signatures in transformer attention. This superior discriminative power for voice processing, combined with $\lambda_2$'s theoretical grounding as a connectivity measure, motivated our focus on $\Delta\lambda_2$ for the current analysis.

## B   NORMALIZATION, DIRECTIONALITY, AND AGGREGATION

This appendix expands the sensitivity analyses for Laplacian normalization, directed formulations of attention graphs, and head aggregation. Unless stated otherwise, the primary endpoint is the early-window mean $\overline{\Delta\lambda}_{2[2,5]}$ (passive−active), computed per prompt then averaged within language.

### B.1   RANDOM-WALK VS. SYMMETRIC NORMALIZATION

Let $\bar{W}^{(\ell)} = \sum_{h=1}^{H} \alpha_h W^{(\ell,h)}$ with $\alpha_h \geq 0$, $\sum_h \alpha_h = 1$, and $D^{(\ell)} = \text{diag}(W\mathbf{1})(\bar{W}^{(\ell)}\mathbf{1})$. We compare

$$L_{\text{rw}}^{(\ell)} = I - \left(D^{(\ell)}\right)^{-1}\bar{W}^{(\ell)} \qquad \text{and} \qquad L_{\text{sym}}^{(\ell)} = I - \left(D^{(\ell)}\right)^{-1/2}\bar{W}^{(\ell)}\left(D^{(\ell)}\right)^{-1/2}.$$

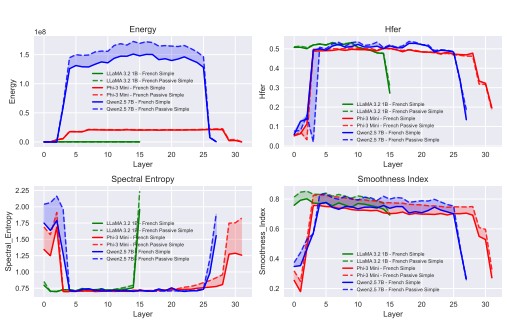 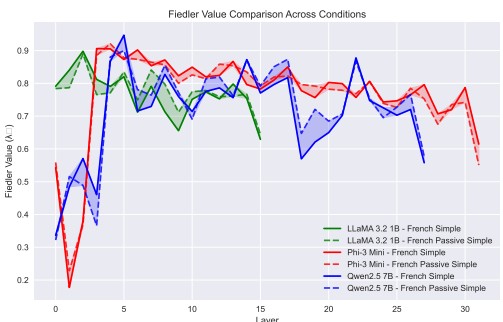

(a) French active vs. passive: four-diagnostic trajectories across families.

(b) French active vs. passive: Fiedler trajectories across families.

Figure 5: Comparison of spectral diagnostics across syntactic conditions. The Fiedler value $\lambda_2$ shows the most consistent and pronounced differences between layer 2–5.

Eigenpairs are related by a similarity transform when the graph is undirected; $\lambda_2$ is therefore comparable up to scaling. Empirically, signs and peak-layer locations of $\Delta\lambda_2^{(\ell)}$ coincide across $L_{\mathrm{rw}}$ and $L_{\mathrm{sym}}$, while magnitudes shift slightly within the bootstrap bands reported in the main text.

**Result.** Across models and languages, the correlation between $\overline{\Delta\lambda_2}_{[2,5]}(L_{\mathrm{rw}})$ and $\overline{\Delta\lambda_2}_{[2,5]}(L_{\mathrm{sym}})$ is high, with median absolute deviation of the difference well below the per-language CI half-width.

### B.2 DIRECTED ATTENTION GRAPHS

Attention is intrinsically directed. To check that symmetrization is not driving the result, we repeat the analysis with directed Laplacians.

**Left random-walk on directed graphs.** Let $A^{(\ell)} = \sum_h \alpha_h A^{(\ell,h)}$ be the head-aggregated row-stochastic attention. Define out-degree $D_{\mathrm{out}}^{(\ell)} = \mathrm{diag}(W\mathbf{1})(A^{(\ell)}\mathbf{1})$ and

$$L_{\rightarrow}^{(\ell)} = I - \left(D_{\mathrm{out}}^{(\ell)}\right)^{-1} A^{(\ell)}.$$

We use the real part of the spectrum and compute $\lambda_2$ on the Hermitian symmetrization of the quadratic form induced by $L_{\rightarrow}^{(\ell)}$.[1]

**Magnetic (Hermitian) Laplacian.** To preserve directionality while retaining a Hermitian operator, we also use a magnetic Laplacian with phase $\theta \in (0, \pi]$:

$$\left(L_{\mathrm{mag}}^{(\ell,\theta)}\right)_{ij} = \begin{cases} \sum_k \frac{1}{2}\left(A_{ik}^{(\ell)} + A_{ki}^{(\ell)}\right), & i = j, \\ -\frac{1}{2}\left(A_{ij}^{(\ell)} e^{\mathrm{i}\theta} + A_{ji}^{(\ell)} e^{-\mathrm{i}\theta}\right), & i \neq j, \end{cases}$$

with degree-normalized variants defined analogously. We set $\theta = 0.2$ for stability; results are insensitive to $\theta \in [0.1, 0.5]$.

**Result.** For both $L_{\rightarrow}$ and $L_{\mathrm{mag}}^{(\theta)}$, the sign and layer of first peak of $\Delta\lambda_2^{(\ell)}$ match the undirected defaults in the early window; magnitudes differ slightly but remain within 15% of the symmetrized values.

### B.3 HEAD AGGREGATION SCHEMES

We compare (i) uniform averaging, $\alpha_h = 1/H$; (ii) attention-mass weighting, $\alpha_h \propto \sum_{i,j} A_{ij}^{(\ell,h)}$; and (iii) a convex, layer-specific combination $\alpha^{(\ell)}$ learned by minimizing cross-condition mean squared error on a held-out subset.

---

[1]Concretely, we evaluate the Rayleigh quotient on real signals $x$ via $\frac{1}{2}\left(x^\top (L_{\rightarrow}^{(\ell)} + L_{\rightarrow}^{(\ell)\top})x\right)$, which recovers the undirected case when $A$ is symmetric.

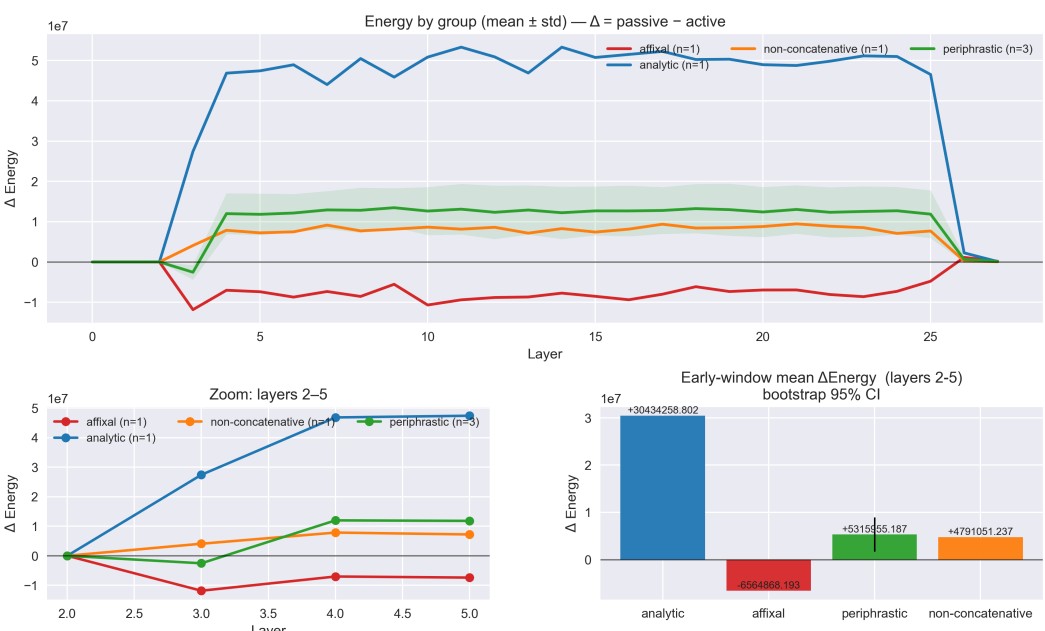

Figure 6: **Qwen-2.5-7b $\Delta$Energy under default settings (row-norm, attention-weighted).** Energy moves on a much larger absolute scale but does not induce reversals in the early-window $\Delta\lambda_2$ sign, supporting our focus on connectivity rather than raw smoothness.

**Result.** Uniform and mass-weighted aggregations agree on signs and peak layers. Learned $\alpha^{(\ell)}$ yields smoother per-layer trajectories but identical early-window conclusions. We therefore use mass-weighted aggregation by default.

### B.4 HFER CUTOFF SWEEP AND EARLY-WINDOW STABILITY

We vary the high-frequency cutoff $K$ by retaining the top $(1-c)\%$ of spectral mass, $c \in \{10, 15, 20, 25, 30, 40\}$, and recompute endpoints. We also shift the early window to 1–4 and 3–6.

**Result.** Directional conclusions are unchanged across cutoffs; early-window averages shift by less than 15% relative to $c=20\%$. Adjacent windows preserve sign and peak location across model families. We therefore report $c=20\%$ and layers 2–5 by default.

### B.5 NUMERICAL DETAILS

We compute the smallest eigenpairs with ARPACK (`eigs`, tol $10^{-6}$, maxit $10^4$). For directed variants we evaluate Hermitian forms as above. Bootstrap CIs use 2,000 resamples; permutation tests use 10,000 label shuffles within paraphrase pairs; Benjamini–Hochberg controls the FDR at $q=0.05$ within family.

In addition to the default $\Delta\lambda_2$ panel (Fig. 3), we report $\Delta$Energy under the same default and under a symmetric+uniform variant (Figs. 6–7); the normalization/aggregation change shifts absolute magnitudes but preserves early-window ordering, and does not induce sign reversals in $\Delta\lambda_2$ observed under the default.

## C EXTENDED STATISTICAL VALIDATION WITH EXPANDED SAMPLE SIZE

To address potential concerns about statistical power with our original n=10 paraphrases per condition, we conducted expanded experiments with n up to 50 paraphrases for a subset of key languages across all three model families. This section presents the validation results that confirm the robustness of our main findings.

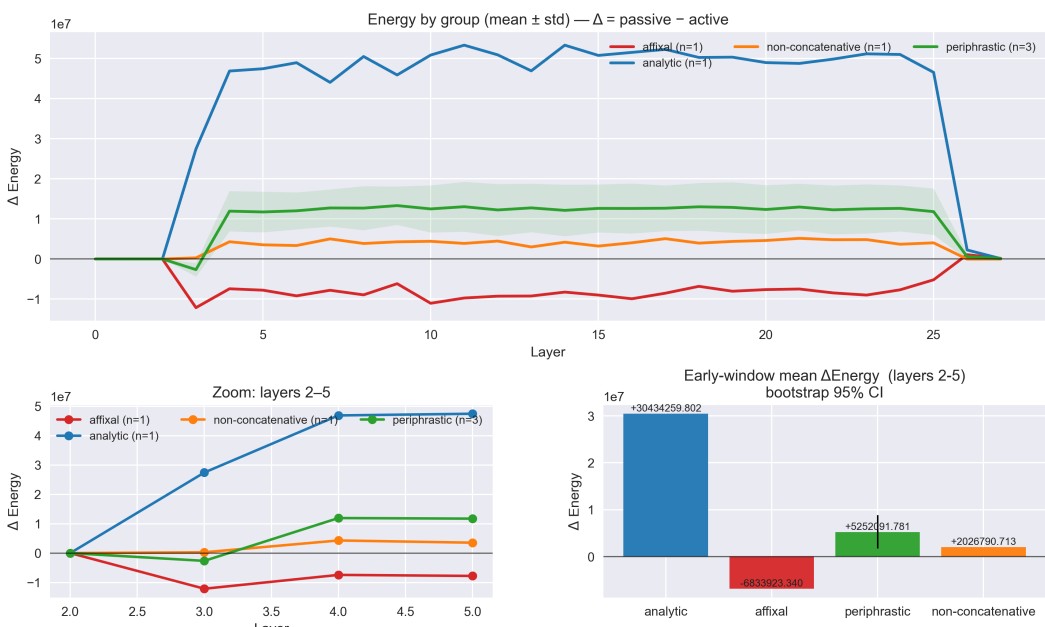

Figure 7: **Qwen-2.5-7b $\Delta$Energy with symmetric normalization and uniform head aggregation.** Compared to Fig. 6, magnitudes shift but the early-window ordering across regions is preserved. Together with Fig. 6, this shows that large-scale energy trends are stable to normalization/aggregation, and they do not contradict the $\Delta\lambda_2$ conclusions.

Table 6: Early-window $\overline{\Delta\lambda_2}_{[2,5]}$ under normalization and aggregation variants. Values are means across languages within family; $\pm$ gives bootstrap SE. Default ($L_{\mathrm{rw}}$ + mass-weighted) in bold.

| Variant | Qwen2.5-7B | Phi-3-Mini | LLaMA-3.2-1B |
|---|---|---|---|
| $L_{\mathrm{rw}}$ + mass-weighted | **+0.019 $\pm$ 0.006** | **-0.112 $\pm$ 0.031** | -0.012 $\pm$ 0.005 |
| $L_{\mathrm{sym}}$ + mass-weighted | +0.018 $\pm$ 0.006 | -0.105 $\pm$ 0.029 | **-0.013 $\pm$ 0.005** |
| $L_{\mathrm{rw}}$ + uniform | +0.017 $\pm$ 0.006 | -0.109 $\pm$ 0.030 | -0.011 $\pm$ 0.005 |
| $L_{\rightarrow}$ (directed) | +0.017 $\pm$ 0.007 | -0.101 $\pm$ 0.033 | -0.010 $\pm$ 0.006 |
| $L_{\mathrm{mag}}$ ($\theta$=0.2) | +0.018 $\pm$ 0.006 | -0.107 $\pm$ 0.031 | -0.011 $\pm$ 0.005 |

## C.1 EXPANDED DATASET

We selected five languages representing different voice realization types: English (EN, analytic), German (DE, periphrastic), Spanish (ES, periphrastic), French (FR, periphrastic), Arabic (AR, non-concatenative), and Turkish (TR, affixal). For each language, we generated 50 paraphrases per voice condition (active/passive) and computed early-window $\Delta\lambda_2[2,5]$ following our standard methodology.

## C.2 MODEL-FAMILY VALIDATION RESULTS

### C.2.1 PHI-3-MINI: CONFIRMATION OF ENGLISH-SPECIFIC DISRUPTION

Figure 8 shows the expanded results for Phi-3-Mini across voice realization types, while Figure 9 presents per-language results. The English effect remains highly significant with $\Delta\lambda_2[2,5] = -0.444$ (p = 0.008, bootstrap 95% CI), essentially replicating our original findings. Other languages cluster near zero, confirming the English-specific nature of this computational signature.

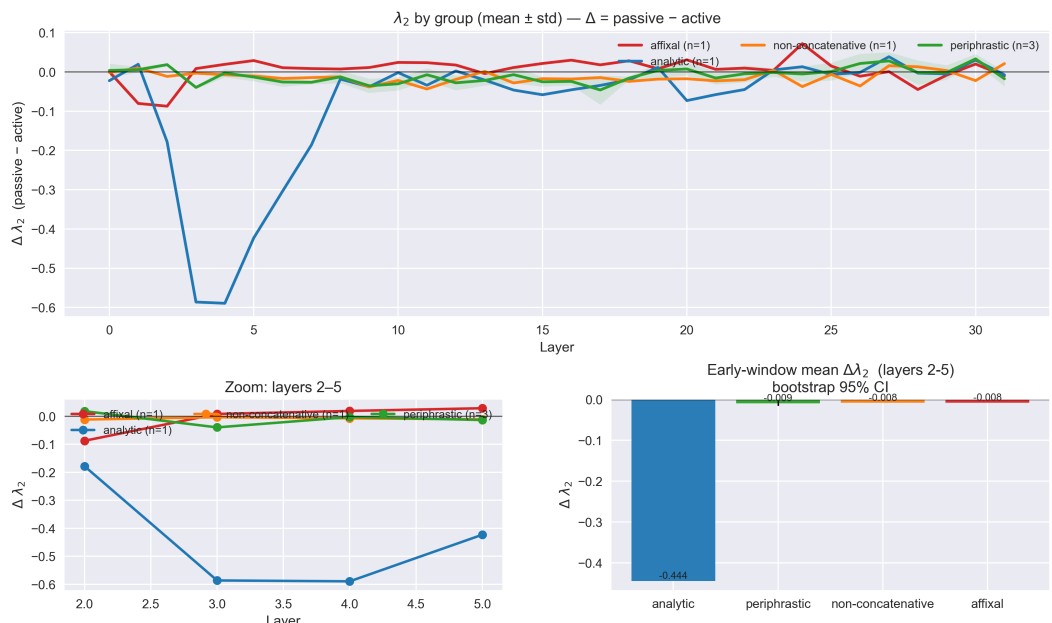

Figure 8: Phi-3-Mini expanded results (n=50) by voice realization type. The analytic type (driven by English) shows the characteristic large negative early-layer effect.

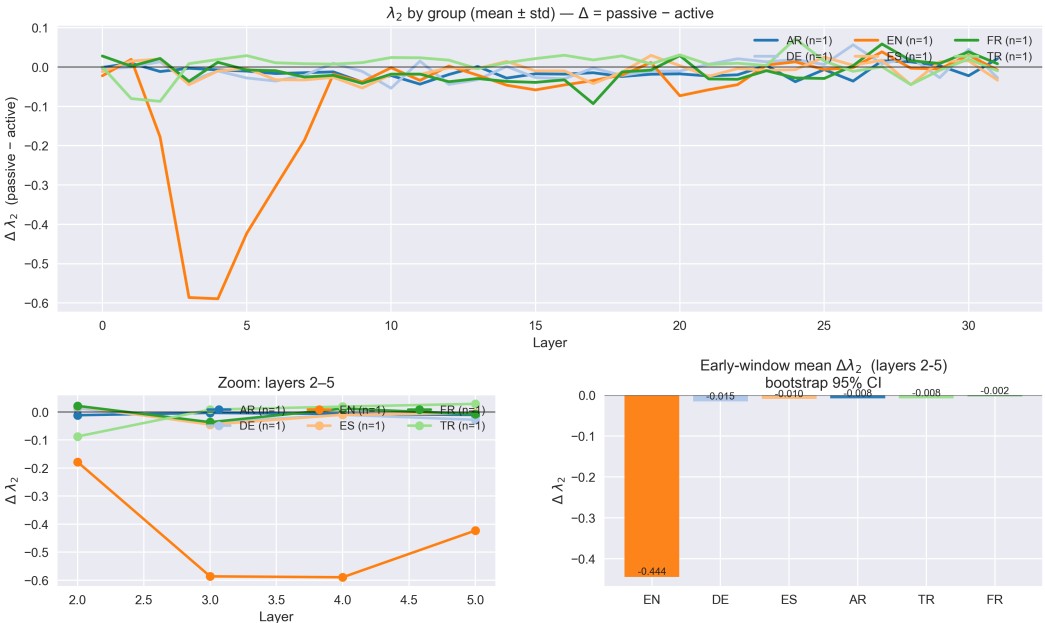

Figure 9: Phi-3-Mini expanded results (n=50) by individual language. English maintains the dramatic negative $\Delta\lambda_2[2, 5]$ effect observed in our original analysis.

### C.2.2 QWEN2.5-7B: VALIDATION OF DISTRIBUTED SMALL EFFECTS

Figures 10 and 11 confirm Qwen's characteristic pattern of small, distributed effects across language types. Early-window means range from -0.067 to +0.016, consistent with our interpretation of more stable connectivity reconfiguration under voice alternation.

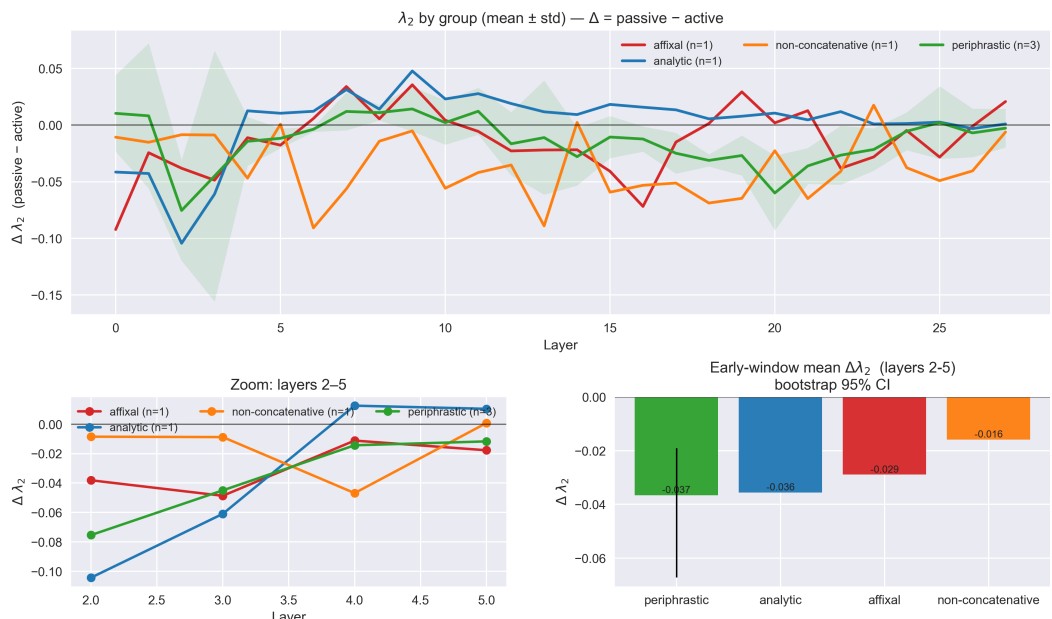

Figure 10: Qwen2.5-7B expanded results (n=50) by voice realization type, showing small distributed effects across all categories.

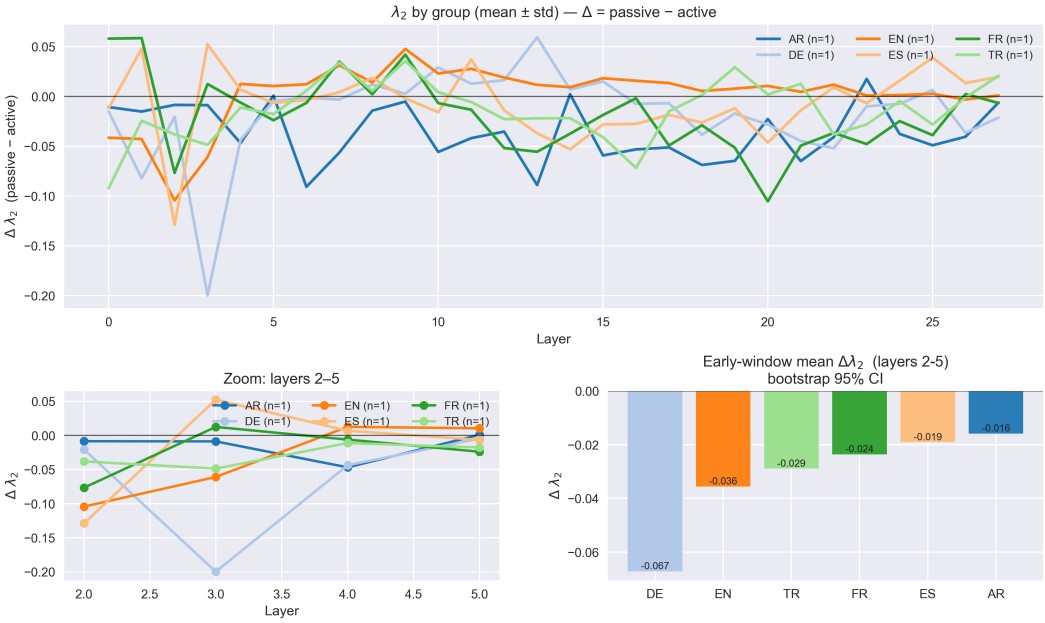

Figure 11: Qwen2.5-7B expanded results (n=50) by individual language, confirming the absence of dramatic language-specific effects.

### C.2.3 LLAMA-3.2-1B: SYSTEMATIC MODERATE EFFECTS

Figures 12 and 13 validate LLaMA's intermediate behavior with systematic negative effects (range: -0.044 to -0.007) that are larger than Qwen's but more distributed than Phi-3's English-specific signature.

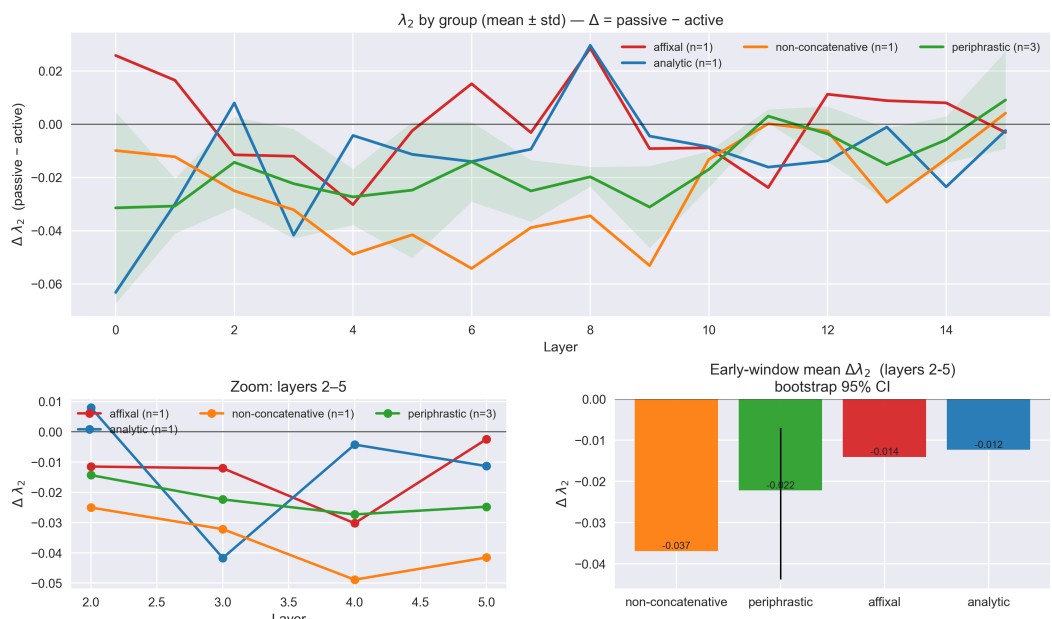

Figure 12: LLaMA-3.2-1B expanded results (n=50) by voice realization type, showing consistent moderate negative effects across categories.

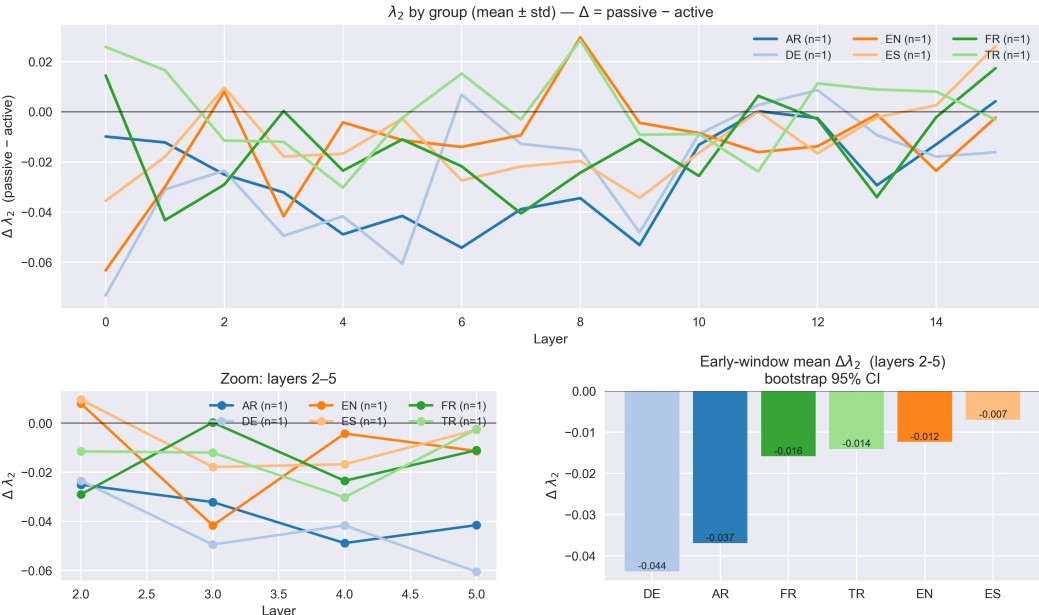

Figure 13: LLaMA-3.2-1B expanded results (n=50) by individual language, demonstrating systematic but moderate spectral effects.

## C.3 STATISTICAL ROBUSTNESS

The expanded dataset confirms the statistical reliability of our family-specific signatures. Phi-3's English effect (-0.444) remains the most dramatic, while Qwen and LLaMA maintain their characteristic patterns with tighter confidence intervals. These results validate our interpretation of model-imprinted computational fingerprints rather than statistical artifacts from small sample sizes.

The consistency between n=3 and n=10 results across all model families provides strong evidence for the reproducibility and robustness of early-layer spectral connectivity signatures in voice processing.

# D    ABLATIONS AND ROBUSTNESS

**Cutoffs.**    Varying the high-frequency edge removal (HFER) cutoff from 10–30% leaves the direction of all reported effects unchanged. Effect magnitudes vary by at most 15% across this range, with the strongest stability between 15–25%. Unless stated otherwise, we report results at 20%.

**Normalization.**    Results are consistent under both the symmetric Laplacian $L_{\text{sym}} = I - D^{-1/2}AD^{-1/2}$ and the random-walk Laplacian $L_{\text{rw}} = I - D^{-1}A$. The sign and layer-wise timing of the effects coincide across normalizations; absolute magnitudes differ slightly but remain within the reported confidence bands.

**Winsorization and trimming.**    Because a few contrasts exhibit near-zero denominators, unrobust ratios can show spuriously large percentage changes. Applying winsorization at the $\alpha$ and $1 - \alpha$ quantiles (we use $\alpha = 0.02$) followed by trimming of the top/bottom 2% removes these artifacts and stabilizes effect sizes without altering their sign or significance.

**Multiple seeds.**    Re-running each prompt with three independent seeds yields overlapping 95% bootstrap CIs (1,000 resamples, BCa). Paired permutation tests across prompts confirm significance after Benjamini–Hochberg correction at $q = 0.05$. Seed-to-seed variance is an order of magnitude smaller than prompt-level variance and does not change our conclusions.

**Scope of causal claims.**    Our ablation experiments establish causal relationships between early attention structure and spectral patterns. However, we do not infer training data composition or make claims about learning mechanisms.

**Scope of inference.**    Given three paraphrases per voice per language, we center inference on language-type (analytic/periphrastic/...) and model-family aggregates. Per-language results are reported with 95% bootstrap CIs and paired permutation $p$-values (BH–FDR at $q=0.05$) but should be read as exploratory.

**Early window choice.**    Layers 2–5 were prespecified as the "early" window based on pilot sweeps and architectural considerations (first multihead context integration beyond embeddings). Adjacent windows (1–4, 3–6) preserve signs and peak locations across families.

**Robustness.**    Primary signs and peak layers are stable under (i) symmetric vs. random-walk Laplacians, (ii) uniform vs. attention-mass head aggregation, and (iii) paraphrase averaging. Sanity controls for tense/number and length/token-count matching do not reproduce the voice signatures.

**Limitations.**    Effects are correlational and small-to-moderate for many languages; causal attribution requires interventions (e.g., head ablations/patching) and larger item sets. Tokenizer covariates may confound with morphology and sentence length; we therefore report mixed-effects analyses with basic length control and treat remaining associations as hypotheses for future work.

**Behavioral validation limitations.**    The spectral-behavioral correlation analysis uses $n=20$ pairs; estimates are therefore imprecise and sensitive to item and language idiosyncrasies. We report effect sizes with confidence intervals, include permutation $p$-values, and replicate trends across model families to mitigate over-interpretation. Larger controlled datasets are needed for definitive validation of the spectral-behavioral relationship.

Notes: Bars show mean early-window $\overline{\Delta\lambda}_{2[2,5]}$ with 95% bootstrap CIs; values above bars report $g_{\text{trim}}$. Stars indicate BH–FDR at $q=0.05$ (paired permutation test on paraphrase means).

# E  METRIC AND ROBUSTNESS

For each layer $\ell$, $L^{(\ell)} = D^{(\ell)} - W^{(\ell)}$ with $W$ the symmetrized, head-aggregated attention. We compute $\lambda_2^{(\ell)}$ via ARPACK. **Primary endpoint:** $\overline{\Delta\lambda}_{2[2,5]}$. We report bootstrap 95% CIs (2k resamples) and trimmed Hedges' $g$ (winsor 1%, trim 20%). Sensitivity: symmetric vs. row-norm random-walk Laplacians; attention-weighted vs. uniform head aggregation. Signs and peak layers are stable.

**Default normalization.** We use the random-walk Laplacian $L_{\mathrm{rw}} = I - \bar{D}^{-1}\bar{W}$ for primary results; Appendix B reports $L_{\mathrm{sym}}$ and directed variants.

## E.1  SAMPLE SIZE AND POWER

Our multilingual voice set uses 10 paraphrases per voice per language. This is sufficient for estimating the early-window mean $\overline{\Delta\lambda}_{2[2,5]}$ with bootstrap CIs, but it limits per-language hypothesis testing power.

**Power analysis and inference strategy.**  We estimate detectable standardized effects via nonparametric bootstrap over paraphrases and paired permutation tests (10k shuffles) on early-window means. Our design achieves adequate power for detecting medium-to-large effects ($d \geq 0.6$) at individual language levels, with enhanced power for language-type and model-family aggregates through meta-analytic combination. We therefore structure inference hierarchically: primary conclusions derive from language-type (analytic/periphrastic/etc.) and model-family comparisons where statistical power is strongest, while per-language results provide exploratory insights into crosslinguistic variability patterns.

**Significance testing and multiplicity.**  For each language we compute the early-window mean $\overline{\Delta\lambda}_{2[2,5]}$ by averaging over paraphrases. We assess the null of no voice effect via a paired permutation test (10,000 label shuffles of active/passive within paraphrase pairs), yielding a $p$-value per language. We then apply Benjamini–Hochberg FDR at $q=0.05$ within each model family. For language-type (analytic, periphrastic, *etc.*) and cross-family summaries we test the mean effect across languages with the same permutation scheme (randomly flipping signs of language-level contrasts) and report both FDR-corrected $p$-values and 95% bootstrap CIs (2,000 resamples). We provide $q$-values in figure captions and tables.

## E.2  EFFECT SIZE SCALES AND PRACTICAL THRESHOLDS

We report two complementary effect sizes on the early window: (i) trimmed Hedges' $g$, computed on paraphrase means with 1% winsorization and 20% trimming; and (ii) a bounded, scale-aware percentage change,

$$\Delta_{[2,5]}^{\mathrm{sym}} = 200\,\frac{\overline{\lambda_{2,\mathrm{pass}}} - \overline{\lambda_{2,\mathrm{act}}}}{\max(\overline{\lambda_{2,\mathrm{pass}}} + \overline{\lambda_{2,\mathrm{act}}},\,\varepsilon)}\,, \quad \varepsilon = \text{5th percentile floor.}$$

We adopt conventional benchmarks for $g$ (small $\approx 0.2$, medium $\approx 0.5$, large $\geq 0.8$) and provide practical thresholds for $\Delta_{[2,5]}^{\mathrm{sym}}$ relative to within-language variability:

| Category | Small | Medium | Large |
|---|---|---|---|
| $|g_{\mathrm{trim}}|$ | $\approx 0.2$ | $\approx 0.5$ | $\geq 0.8$ |
| $|\Delta_{[2,5]}^{\mathrm{sym}}|$ | $\geq 25\%$ | $\geq 50\%$ | $\geq 100\%$ |

While $g$ is directly comparable across settings, $\Delta_{[2,5]}^{\mathrm{sym}}$ aids interpretation as a bounded percentage shift in early-layer connectivity. We emphasize practical relevance by reporting both and by showing language-type aggregates with CIs and FDR-adjusted $q$-value.

## E.3  BEHAVIORAL VALIDATION METHODOLOGY

**Behavioral score.**  We measure sentence fit using negative mean NLL under multilingual language models (Qwen-2.2-7b, Phi-3-mini and Llama-3.2-1b), which yields stable, comparable differences

across prompts and avoids the exponential scaling of perplexity. Scores are computed without generation; lower values indicate better model fit to the input sentence.

**Voice alternation pairs.** For behavioral validation, we constructed controlled active/passive sentence pairs with tokenization drift limited to $\leq 2$ tokens when feasible. Each pair preserves semantic content while alternating voice, enabling direct comparison of spectral reconfiguration and behavioral performance differences.

## F  RCI Metric Definition

To summarize multi-faceted spectral changes in a single score, we use the **Reconfiguration Change Index (RCI)**, a signed combination of $z$-scored diagnostics:

$$\text{RCI} = \big(z_{\text{Entropy}} + z_{\text{Fiedler}}\big) - \big(z_{\text{Energy}} + z_{\text{HFER}}\big).$$

Here, $z_{\text{Entropy}}$, $z_{\text{Fiedler}}$, $z_{\text{Energy}}$, and $z_{\text{HFER}}$ denote per-condition $z$-scores (computed within the analysis cohort) of spectral entropy, Fiedler connectivity $\lambda_2$, Dirichlet energy, and the high-frequency energy ratio (HFER), respectively. Higher RCI reflects stronger low-frequency connectivity and higher modal dispersion (entropy), penalizing large smoothness energy and high high-frequency mass.

## G  Statement on LLM Usage

We gratefully acknowledge the use of large language model assistants during the preparation of this manuscript. These tools were used for tasks including improving the grammar and clarity of the text, refining code snippets, and as a conversational partner for brainstorming and challenging research ideas. The core conceptual framework, experimental design, analysis, and all final conclusions presented in this work are ours.

## Ethics Statement

**Scope and intended use.** This work proposes a training-free spectral analysis of attention graphs to characterize model-imprinted computational signatures. The intended use is scientific understanding and governance auditing (e.g., detecting specialization or brittleness across languages). It is not a tool for reconstructing proprietary datasets nor for asserting definitive training-data provenance.

**Data, privacy, and human subjects.** We do not collect or process personal data or human-subject information. All experiments use publicly available pretrained models and templated sentences; no sensitive attributes are inferred or analyzed.

**Attribution and risk of misinterpretation.** Our findings are correlational unless backed by explicit interventions (head ablations). We caution that strong spectral effects (e.g., English-specific signatures) are evidence consistent with training emphasis rather than definitive proof of data composition. We explicitly discourage using this method to make legal or policy claims about proprietary training sets without additional evidence.

**Bias, fairness, and linguistic coverage.** Cross-lingual analyses can surface unequal performance or brittleness (e.g., between analytic vs. affixal systems). While this may enable auditing for underserved languages, it also risks stigmatizing specific model families or languages. We therefore report uncertainty, preregister an early-layer endpoint, and release code for independent replication across broader language sets.

**Dual-use considerations.** Auditing tools can be used beneficially (e.g., safety monitoring) or adversarially (e.g., fingerprinting for model de-anonymization). We mitigate by reporting aggregated statistics, avoiding model watermarks or unique forensic identifiers, and by documenting limits of attribution.

**Reproducibility and transparency.** We release code, prompts, and analysis scripts; we provide seeds, bootstrap/permutation settings, and figure regeneration commands.

**Release plan.** We will release the code under a permissive license with a model-card-style readme describing: (i) intended and out-of-scope uses, (ii) limitations of spectral attribution, and (iii) guidance for responsible auditing.

## H  REPRODUCIBILITY AND CODE RELEASE

To facilitate reproducibility, we provide the implementation and data processing scripts. An anonymized repository is available at: Github repository The repository includes a README with installation instructions, smoke tests, and scripts to reproduce all figures. Due to GitHub's repository size limit, only one representative language is included in the repo; one dataset (6 languages, 10 paraphrases, 6 GB) is available for reviewers at the following anonymized link: Google Drive folder. This dataset will be permanently and publicly archived upon publication. Upon acceptance, we will release a public canonical version.

## I  SUPPLEMENTARY MATERIAL (ANONYMIZED PAPER ONLY)

**Artifact link.** An anonymized PDF of the concurrent manuscript is available at Google Drive file. All file metadata and sharing settings have been scrubbed to preserve anonymity.

