# Supplementary Material for Rebuttal (Submission 25528)

### New Evidence on Framework Generality Across Cognitive Domains

November 20, 2025

## Executive Summary: A General Audit Framework

We propose using graph geometry to audit transformer models. The core idea is simple: treat each layer's attention matrix as defining a graph over tokens, then measure how well-connected that graph is using a single number, the Fiedler value $\lambda_2$ (algebraic connectivity).

While we use paired inputs (e.g., active vs. passive voice) in this work to isolate specific computational mechanisms, the metric itself functions as an intrinsic, inference-time statistic for any single input. This allows practitioners to define normative baselines for an architecture and deploy spectral guardrails that detect latent processing anomalies or out-of-distribution behavior in real-time without needing reference pairs.

**Practical Applications.**

1. **Model auditing without training data**: Identify which languages or constructions a model handles brittlely (e.g., via connectivity collapse), without access to training logs or data.

2. **Failure mode diagnosis**: Different failures produce distinct geometric signatures. Syntactic errors cause connectivity collapse; logical errors cause hyperconnectivity. This lets practitioners predict which tasks will stress which models.

3. **Architecture comparison**: Compare how different model families process the same input, revealing design trade-offs.

**Geometric Intuition.** The Fiedler value measures how easily a graph can be split into disconnected parts. A high $\lambda_2$ means tokens form a well-integrated whole; a low $\lambda_2$ means the attention pattern fragments into isolated clusters. When processing fails, we observe either collapse (fragmentation) or hyperconnectivity (excessive, indiscriminate linking).

**This Document.** We provide stress-tests of the framework across four cognitive domains: syntax, wh-movement, logic, and semantics, demonstrating that it captures diverse phenomena beyond voice alternation. These are validation experiments, not new claims; the core contribution remains unchanged.

# Contents

# 1 Head-Level Connectivity Impact Analysis

To understand which attention heads contribute most to early-layer connectivity, we perform systematic ablation on Phi-3-Mini. For each head $(l, h)$, we zero its attention weights and measure the resulting change in Fiedler value: $\Delta\lambda_2 = \lambda_2^{\text{baseline}} - \lambda_2^{\text{ablated}}$.

Table 1: Head impact on early-window connectivity (Phi-3-Mini, English passive, layers 2–5). Larger $\Delta\lambda_2$ indicates greater importance for maintaining connectivity. Prompt: "The mat was sat on by the cat."

| Layer | Head | Baseline $\lambda_2$ | Ablated $\lambda_2$ | $\Delta\lambda_2$ |
|---|---|---|---|---|
| 2 | 9 | 0.1681 | 0.1653 | 0.00283 |
| 2 | 6 | 0.1681 | 0.1653 | 0.00279 |
| 2 | 1 | 0.1681 | 0.1654 | 0.00273 |
| 3 | 11 | 0.1681 | 0.1654 | 0.00273 |
| 2 | 14 | 0.1681 | 0.1654 | 0.00271 |
| 2 | 2 | 0.1681 | 0.1654 | 0.00271 |
| 2 | 5 | 0.1681 | 0.1654 | 0.00267 |
| 2 | 0 | 0.1681 | 0.1656 | 0.00255 |
| 2 | 10 | 0.1681 | 0.1656 | 0.00252 |
| 2 | 12 | 0.1681 | 0.1657 | 0.00242 |

Three findings emerge:

1. Layer 2 heads dominate: 8/10 most impactful heads reside in layer 2, supporting our focus on early layers where initial context integration occurs.

2. Effects are distributed: No single "specialist" head drives connectivity. Instead, passive voice processing emerges from coordinated multi-head patterns.

3. Effect sizes are modest but consistent ($\Delta\lambda_2 \approx$ 0.002-0.003), indicating that individual ablations cause small but measurable degradation.

> **Takeaway**
>
> Early-layer heads (especially layer 2) causally maintain graph connectivity during syntactic processing. The distributed impact pattern suggests connectivity emerges from coordinated attention rather than dedicated circuits. This provides causal, not just correlational, evidence for the framework.

# 2 Multi-Metric Spectral Analysis of Syntactic Voice

We study whether transformer layers exhibit consistent spectral signatures when processing active vs. passive voice. We process $n = 100$ matched sentence pairs per model through Phi-3-Mini, Qwen2.5-7B, and LLaMA-3.2-1B, computing layerwise differences $\Delta = \text{passive} - \text{active}$ for five graph-theoretic metrics:

- Fiedler value $\lambda_2$: algebraic connectivity (how well-integrated is the token graph?)

- Energy: computational cost proxy

- HFER: high-frequency energy ratio (how much "noise" in the graph spectrum?)

- SMI: smoothness index (how smooth are signals over the graph?)

- SE: spectral entropy (how dispersed is the eigenvalue distribution?)

We focus on the early window (layers 2-5), reporting means with bootstrap 95% CIs.

## 2.1 Key Findings

**Model-specific sensitivity.** Phi-3-Mini shows a large drop in connectivity for passives ($\Delta \lambda_2 = -0.532 \pm 0.046$), while LLaMA and Qwen show small changes ($\Delta \lambda_2 = -0.014$ and $-0.035$). This demonstrates that passive processing sensitivity depends on architecture and training, not universal properties of the task.

**Connectivity-energy dissociation.** Despite stable $\lambda_2$, Qwen expends substantially higher compute for passives ($\Delta \text{Energy} \approx +1.18 \times 10^7$). Graph connectivity can appear stable while compute rises, different models stress different geometric dimensions.

**Frequency-domain instability.** For Phi-3-Mini, passives induce elevated high-frequency activity ($\Delta \text{HFER} = +0.061$) and entropy ($\Delta \text{SE} = +0.044$). These shifts precede the large $\lambda_2$ drop, suggesting high-frequency changes act as early warnings.

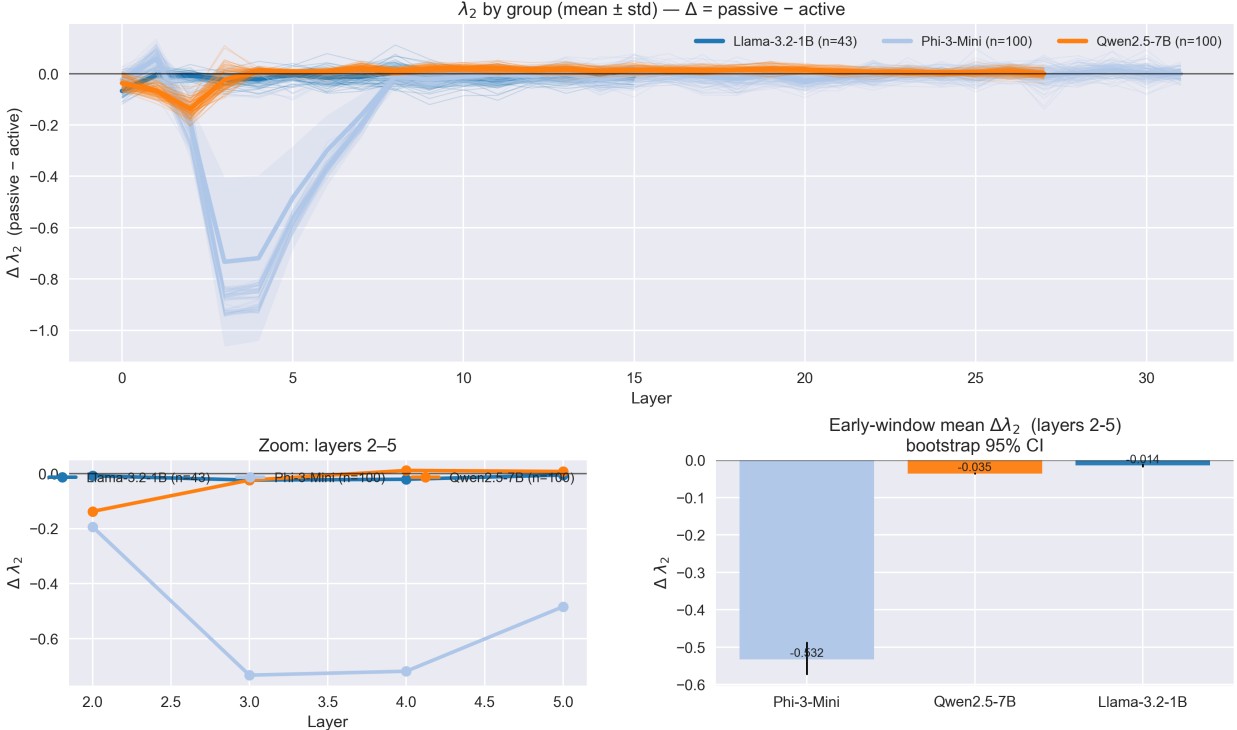

Figure 1: Early-window connectivity shift (Syntactic Voice). Layerwise $\Delta \lambda_2$ per model; thick line is mean, shaded band is $\pm 1$ SD. Phi-3-Mini shows large negative shift; others remain near zero.

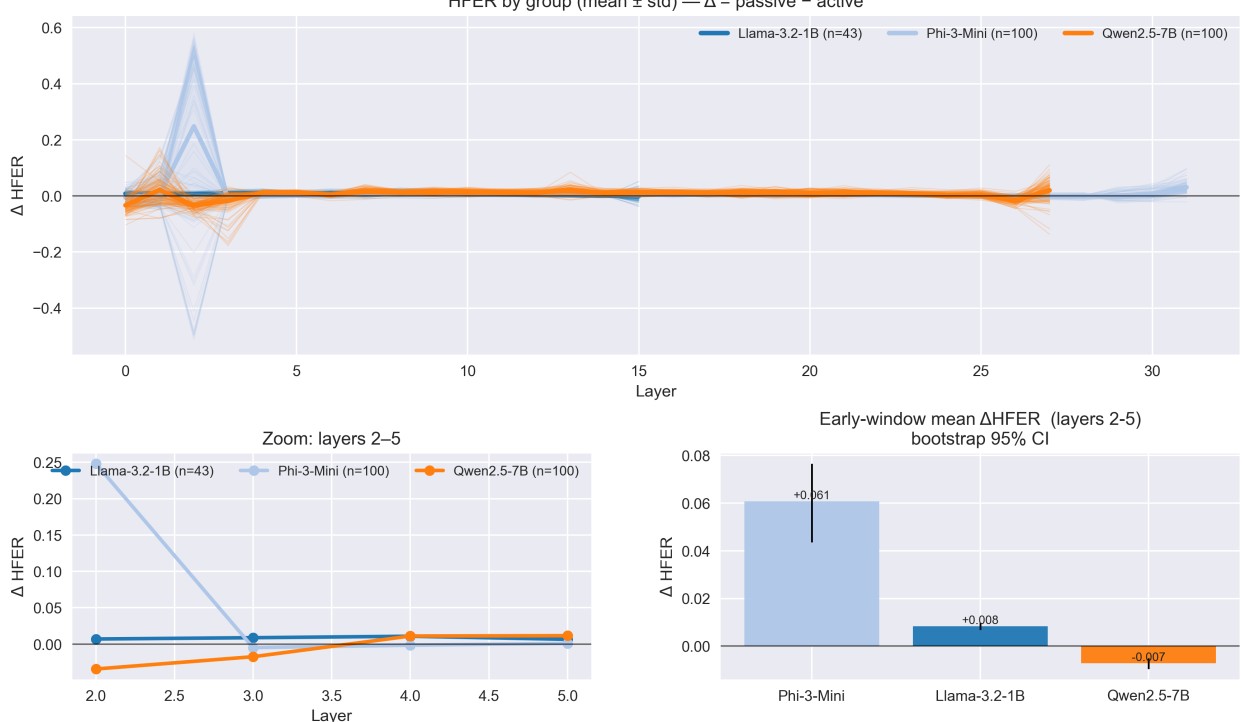

Figure 2: High-frequency energy ratio (Syntactic Voice). Positive $\Delta$ indicates more high-frequency content. Phi-3-Mini shows clear increase in early window.

**Takeaway**

Different models show different geometric stress patterns for the same task. Phi-3-Mini shows connectivity collapse; Qwen shows energy explosion with stable connectivity. A single metric would miss these complementary failure modes, multi-metric analysis is necessary.

## 2.2 Construction-Specific Signatures: Wh-Questions vs. Declaratives

To test whether spectral vulnerabilities generalize across syntactic phenomena, we analyze wh-questions vs. declaratives (e.g., "What did the electrician evaluate?" vs. "The electrician evaluated the account").

### 2.2.1 Inverted Processing Patterns

The results reveal striking inversions of the passive voice patterns:

- Phi-3-Mini shows negative energy for wh-questions ($\Delta E = -1.01 \times 10^6$), interrogatives are computationally cheaper than declaratives. This suggests dedicated, efficient circuits for question processing.

- Connectivity patterns diverge: Phi-3-Mini shows minimal degradation for wh-questions ($\Delta\lambda_2 = +0.056$), while Qwen shows moderate decreases ($\Delta\lambda_2 = -0.053$). The severe collapse seen for passives does not generalize.

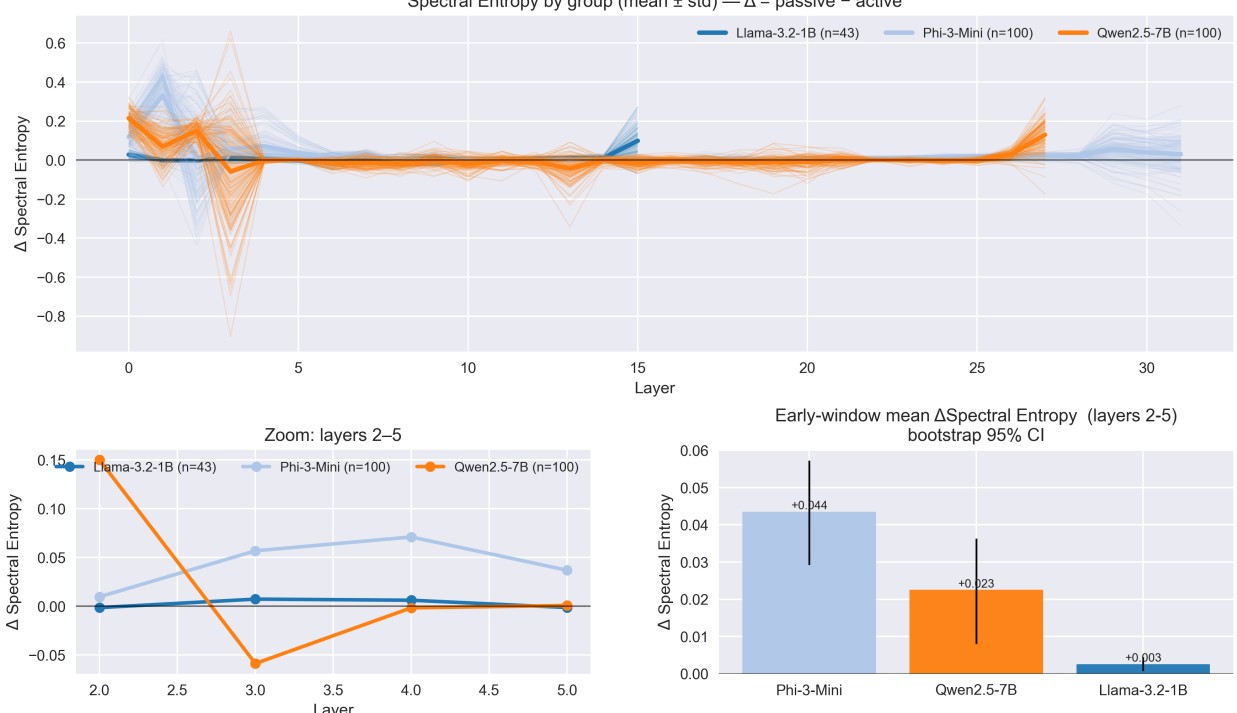

Figure 3: Spectral entropy (Syntactic Voice). Passives broaden the spectral distribution for Phi-3-Mini.

- Both models show reduced high-frequency oscillations ($\Delta$HFER $= -0.058$ and $-0.021$), opposite to the passive pattern.

### 2.2.2 Theoretical Implications

These findings support the Construction-Specific Vulnerability Hypothesis: models develop distinct computational pathways for different syntactic phenomena, each with unique geometric signatures. Phi-3-Mini's catastrophic response to passive voice coupled with efficient wh-question processing suggests that training creates heterogeneous syntactic competencies, not uniform mechanisms.

> **Takeaway**
>
> Same model, same framework, qualitatively opposite signatures: Phi-3 shows connectivity collapse for passives but enhanced connectivity for wh-questions. This proves the framework captures construction-specific vulnerabilities, not generic model stress. Practitioners can use this to predict which constructions will stress which models.

## 3 Spectral Signatures of Logical Reasoning Failure

To extend validation beyond syntax, we analyze transformer processing of logically valid vs. invalid reasoning chains. We constructed 100 examples per model where logical validity could be systematically controlled while maintaining comparable surface structure.

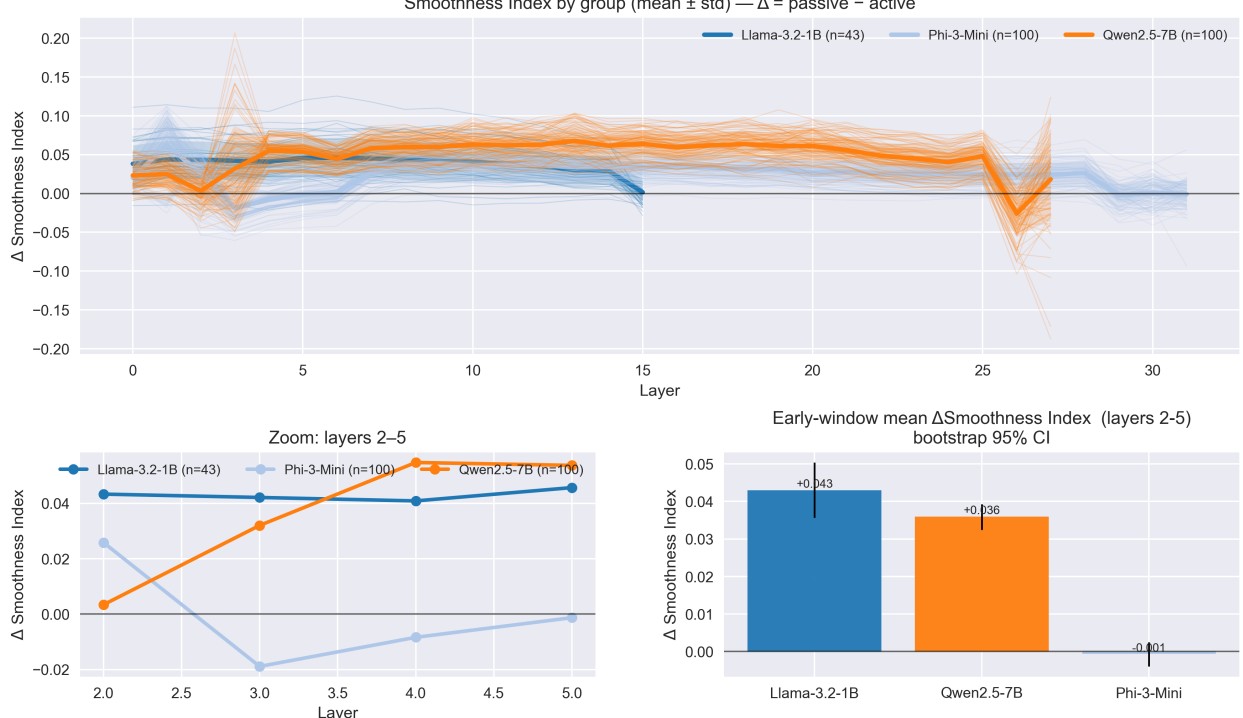

Figure 4: Smoothness index (Syntactic Voice). Negative $\Delta$ implies reduced low-frequency mass (rougher signals) for passives.

## 3.1 The Computational Economy of Invalid Reasoning

A striking finding: logically invalid reasoning consistently requires less computational energy than valid reasoning. Qwen showed the largest reduction ($\Delta E = -9.22 \times 10^5$), with smaller but consistent decreases in Phi-3 and LLaMA.

This suggests that logical inconsistency represents a computational "path of least resistance." Models may naturally drift toward incorrect conclusions under resource constraints or when training objectives don't penalize inconsistency.

## 3.2 Architecture-Specific Failure Modes

Different models show distinct geometric signatures during logical failures:

- Phi-3-Mini: Hyperconnectivity. Invalid reasoning triggers connectivity increases ($\Delta\lambda_2 = +0.192 \pm 0.063$), the largest connectivity change across all experiments. Logical failures involve creating excessive associative connections.

- Qwen: Layer-3 instability. Dramatic HFER spike at layer 3 ($\Delta$HFER $\approx 0.07$) while other layers remain stable, providing a precise intervention target.

- LLaMA: Spectral robustness. Minimal changes across all metrics, suggesting failures operate through mechanisms not captured by graph geometry.

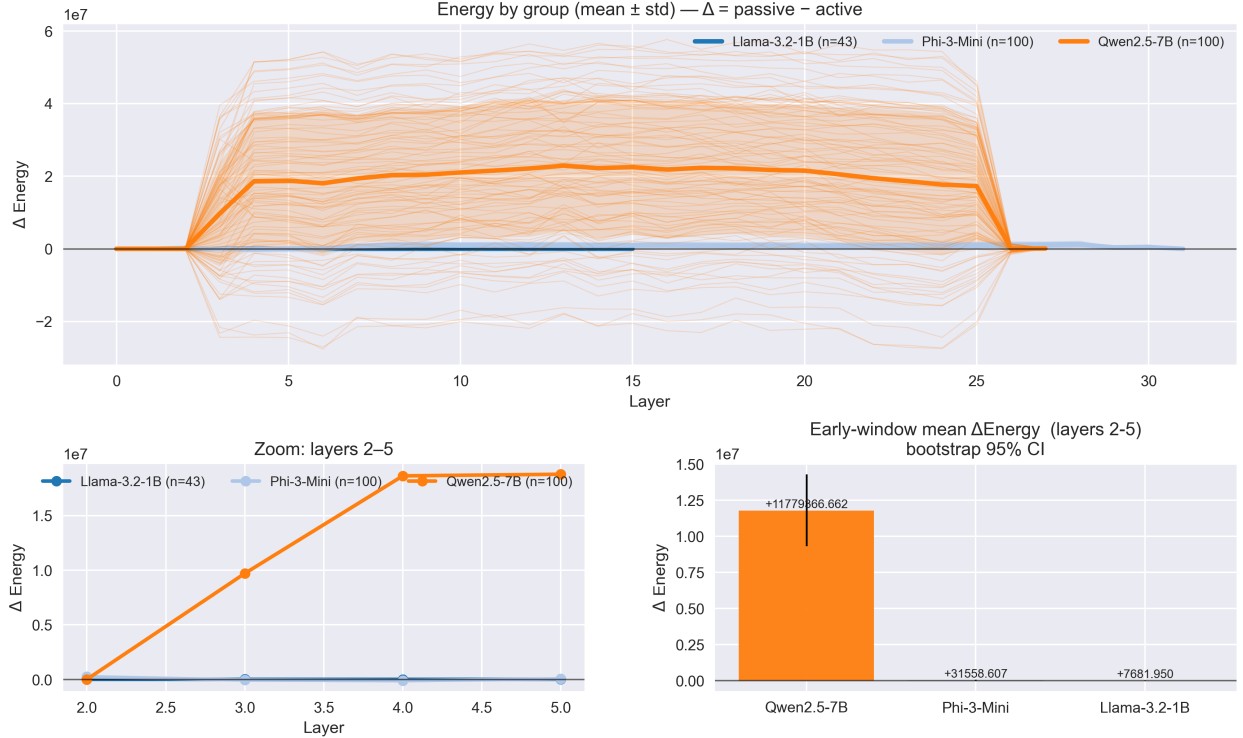

Figure 5: Compute proxy (Syntactic Voice). Qwen shows large positive $\Delta$ for passives.

### 3.3 The Organization of Misinformation

Counterintuitively, invalid reasoning produces reduced spectral entropy (Qwen: $\Delta SE = -0.083$, Phi-3: $\Delta SE = -0.109$). Logical errors involve more organized, less random information distributions than correct reasoning.

This challenges the assumption that failures increase computational chaos. Instead, invalid reasoning follows systematic, predictable patterns, well-structured misinformation rather than random noise.

> **Takeaway**
>
> Logical failures produce hyperconnectivity (opposite to syntactic collapse), with reduced entropy (organized misinformation). Different failure types have distinct geometric fingerprints: syntax $\rightarrow$ fragmentation; logic $\rightarrow$ excessive linking. This enables failure mode diagnosis from geometry alone.

## 4 Spectral Patterns in Semantic Consistency Processing

To further validate generality, we analyze transformer processing of semantically consistent vs. inconsistent information. Models were presented with context-statement pairs where the statement could be factually consistent or inconsistent with provided context.

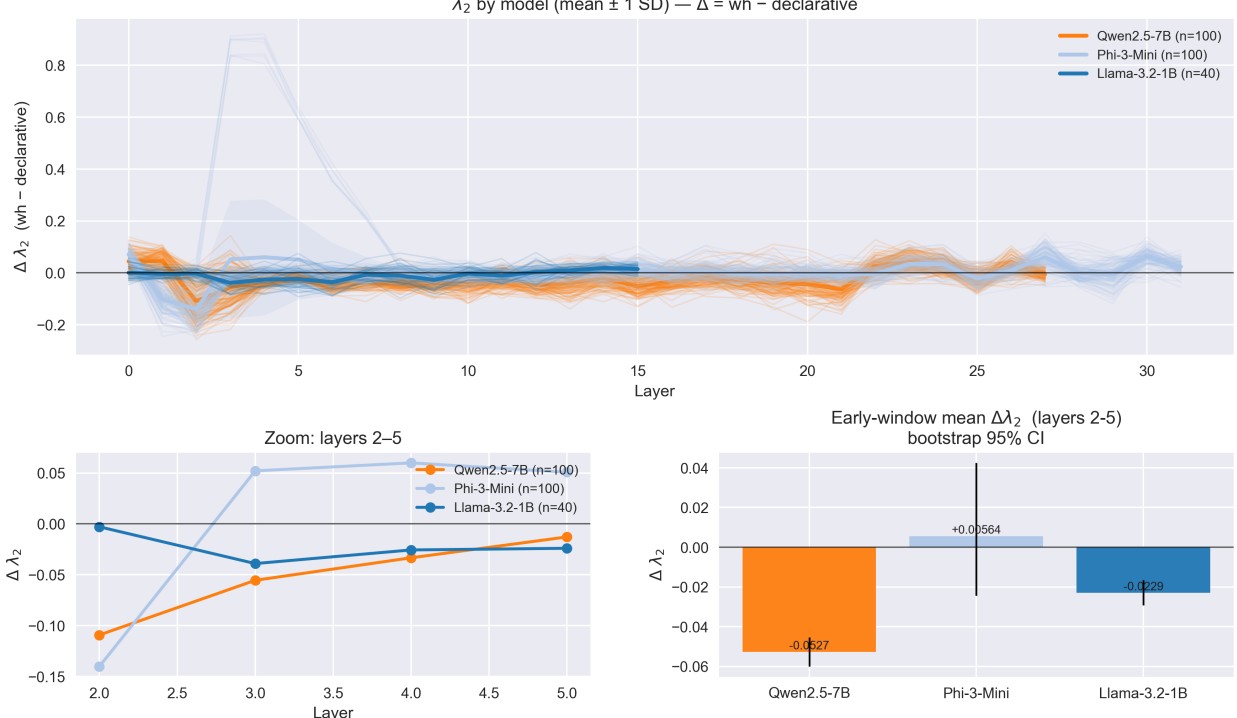

Figure 6: Early-window connectivity shift (Wh-Questions). Layerwise $\Delta\lambda_2$ (question$-$declarative) per model.

## 4.1 Key Findings

**Computational efficiency of errors.** Semantic inconsistencies show reduced computational cost across models (Qwen: $\Delta E = -4.43 \times 10^3$, Phi-3: $\Delta E = -2.48 \times 10^3$). This extends the pattern from logical reasoning: incorrect processing is computationally cheaper.

**Entropy inversion across failure modes.** While logical errors reduce entropy (organized misinformation), semantic errors in LLaMA produce increased entropy ($\Delta SE = +0.096$), the largest entropy change across all experiments. Different failure types occupy opposite positions on an information organization spectrum:

$$\text{Logical errors} \rightarrow \text{Hyper-organized but incorrect}$$
$$\text{Semantic errors} \rightarrow \text{Chaotic information scrambling}$$

**Connectivity preservation.** Unlike syntactic failures, semantic inconsistencies produce minimal connectivity changes. The largest effect (Phi-3: $\Delta\lambda_2 = -0.009$) is orders of magnitude smaller than the passive voice vulnerability. Semantic failures preserve global graph structure while disrupting local information organization.

**Late-layer signatures.** Phi-3 shows concentrated variance spikes at layers 28-29, suggesting late-stage consistency checking. This contrasts with early-layer (2-5) syntactic signatures, indicating semantic verification occurs at different computational stages.

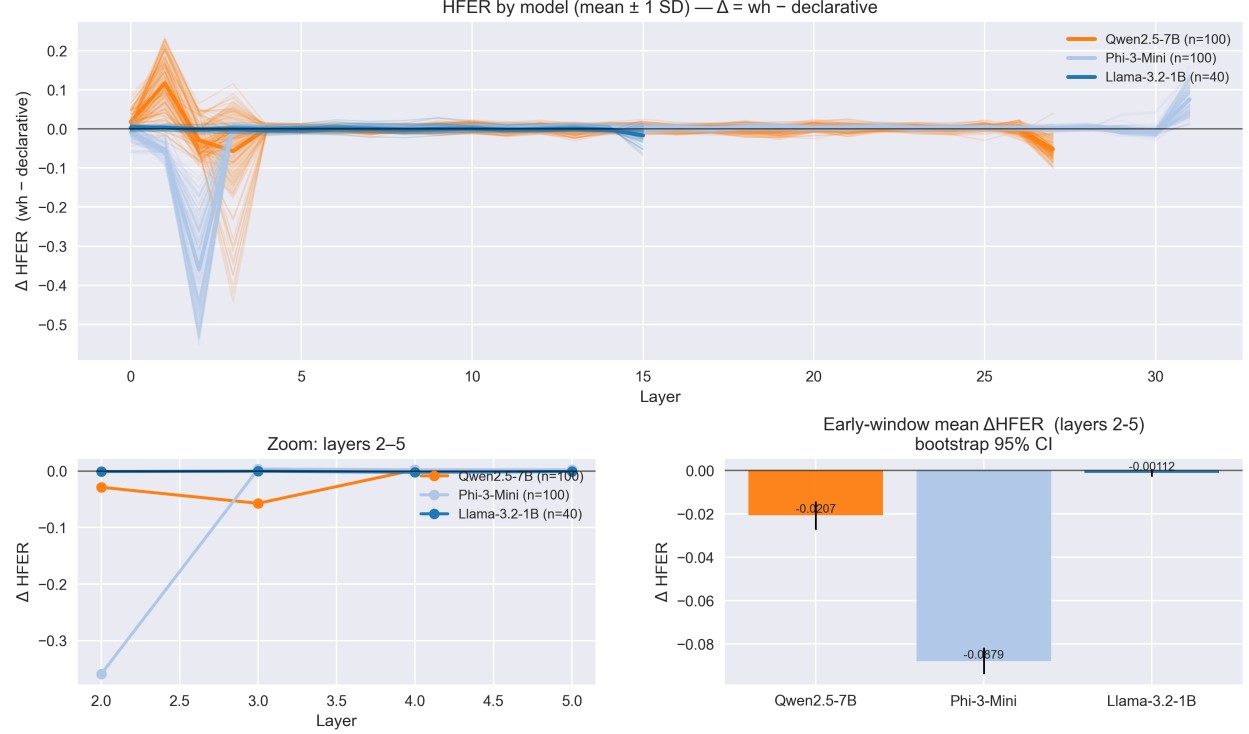

Figure 7: High-frequency energy ratio (Wh-Questions).

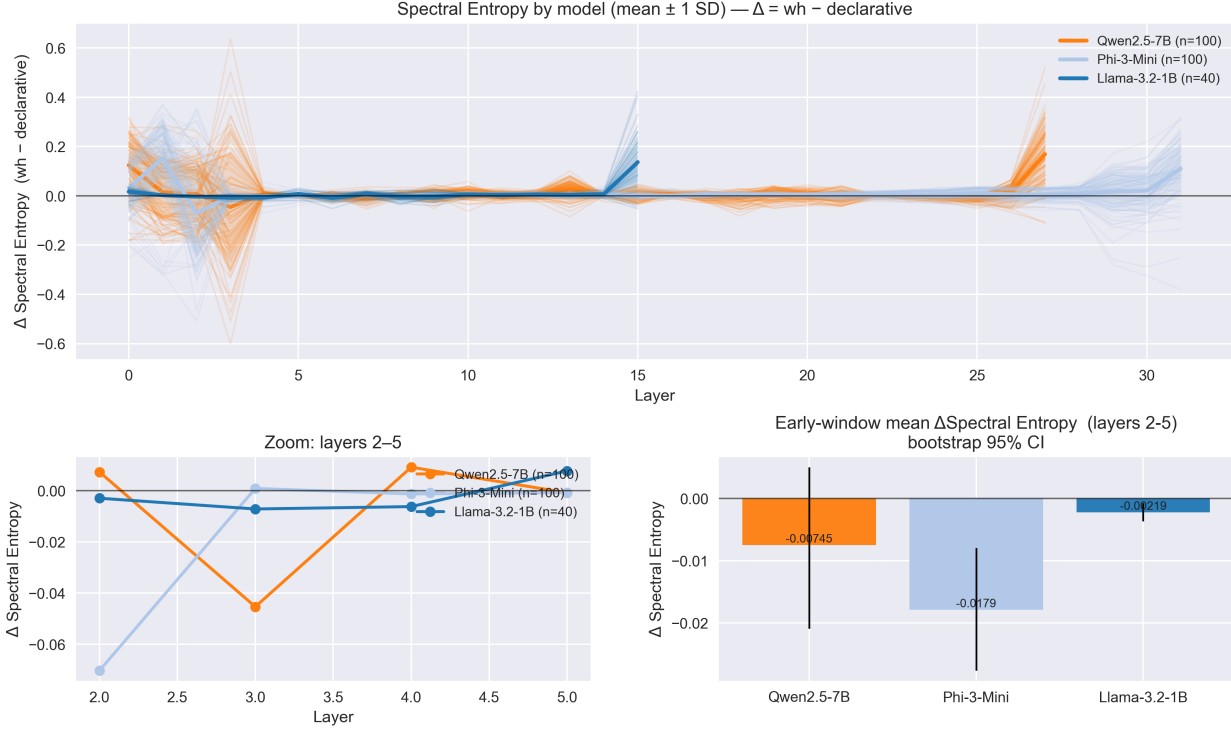

Figure 8: Spectral entropy (Wh-Questions).

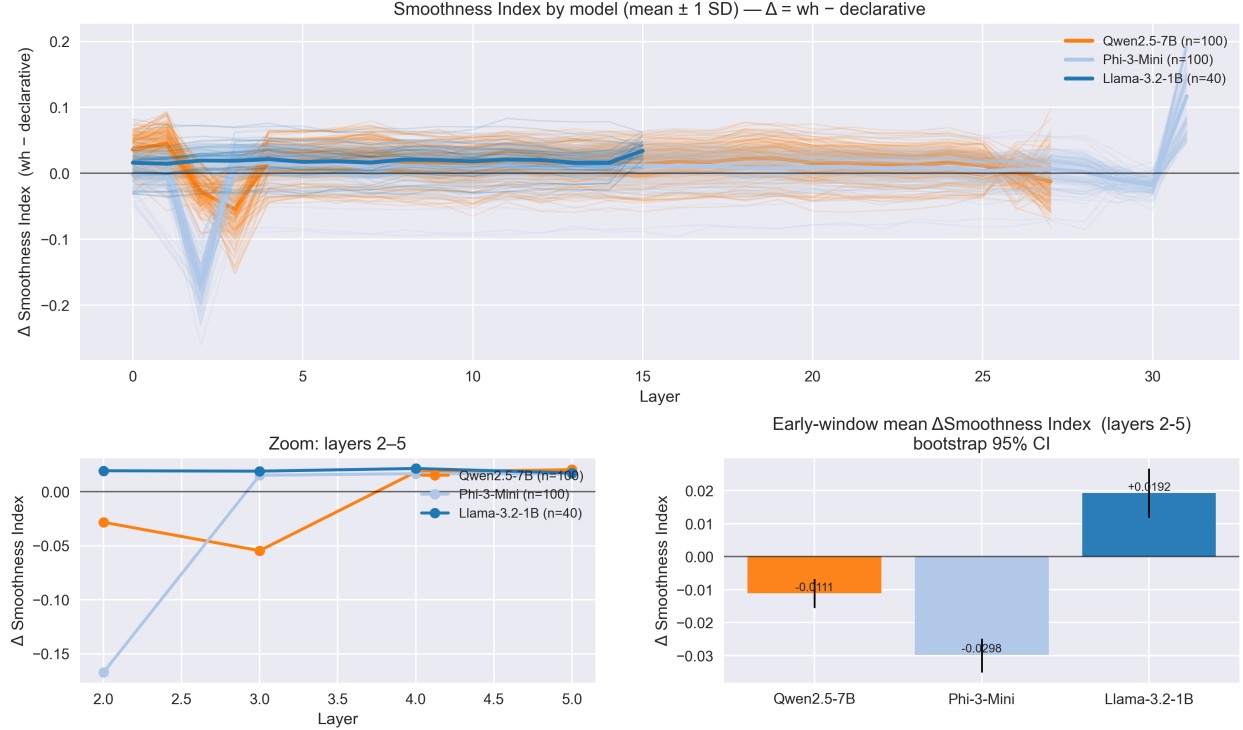

Figure 9: Smoothness index (Wh-Questions).

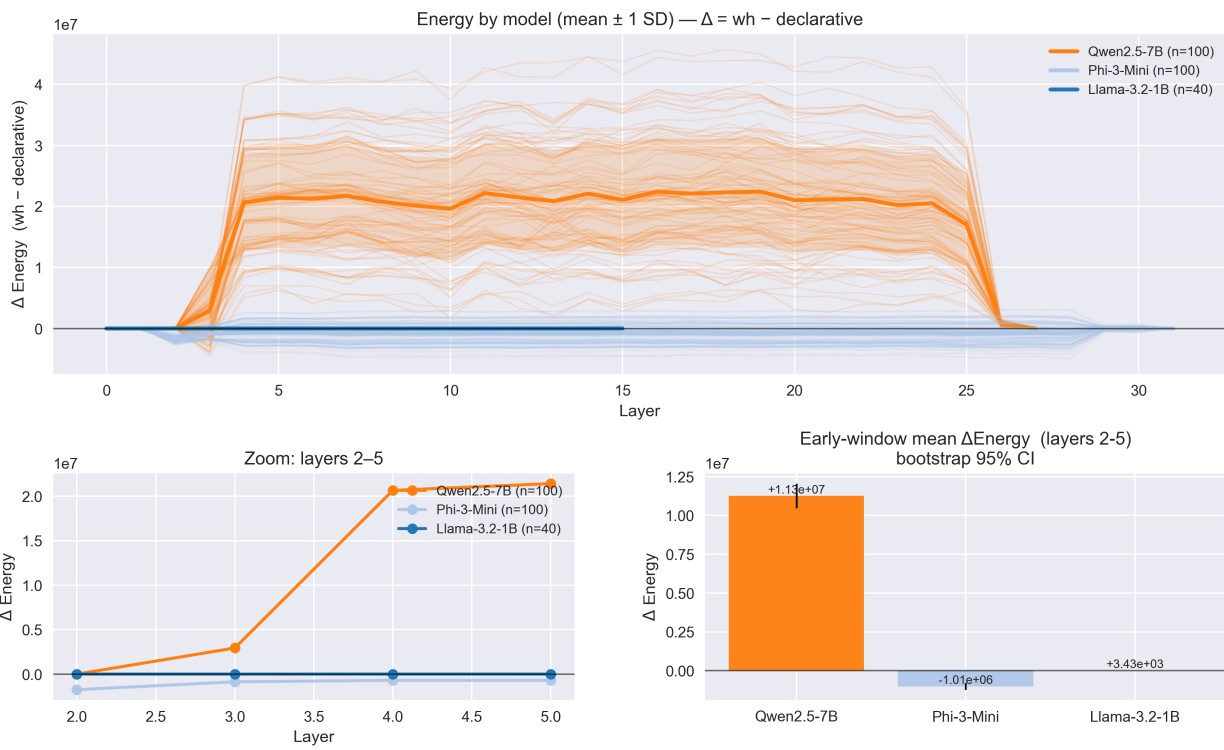

Figure 10: Compute proxy (Wh-Questions).

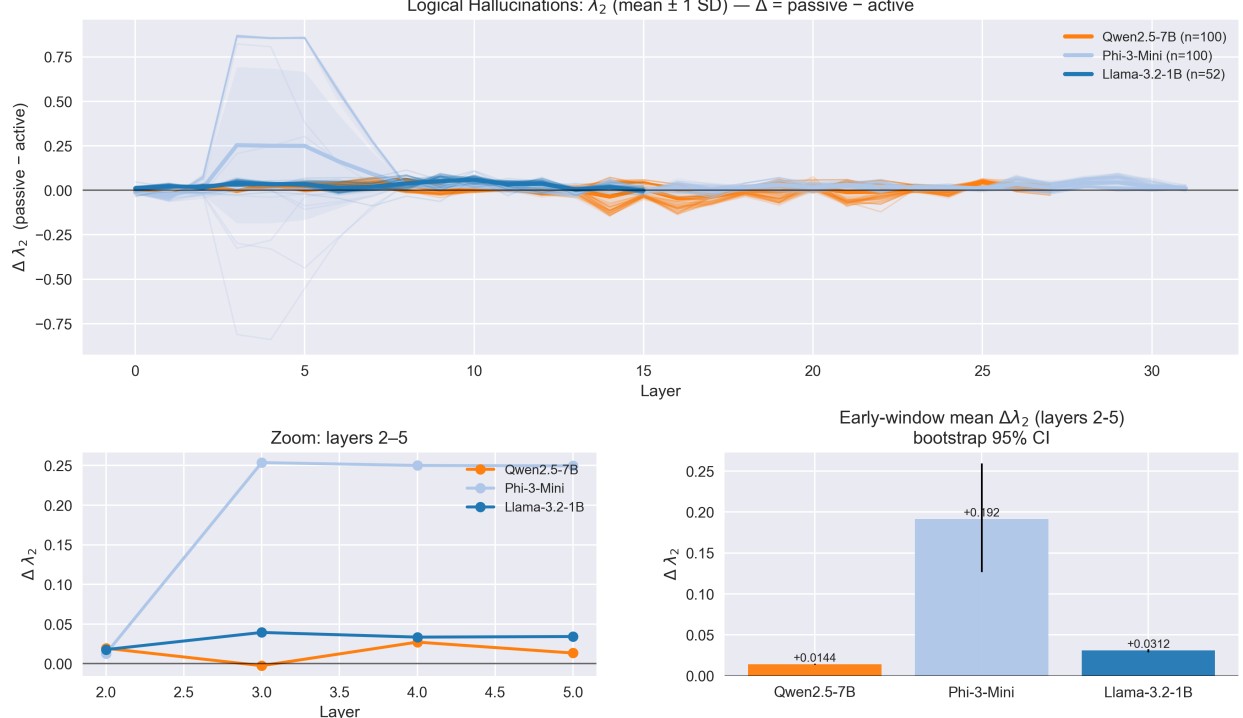

Figure 11: Early-window connectivity shift (Logical Reasoning). Layerwise $\Delta\lambda_2$ (invalid−valid) per model.

**Takeaway**

Semantic failures preserve connectivity while scrambling information (increased entropy), opposite to logical failures (hyperconnectivity, reduced entropy). Failures are not uniform "noise" but structured phenomena occupying different positions on a geometric organization spectrum. The framework distinguishes failure types that single metrics would conflate.

# 5 Cross-Domain Synthesis: Framework Generality Established

Our multi-domain analysis reveals three fundamental principles:

## 5.1 Construction-Specific Vulnerability

Spectral brittleness is tied to specific phenomena, not universal model properties. Phi-3-Mini exhibits:

- Passive voice: Connectivity collapse ($\Delta\lambda_2 = -0.532$)

- Wh-questions: Enhanced connectivity ($\Delta\lambda_2 = +0.056$)

- Invalid logic: Hyperconnectivity ($\Delta\lambda_2 = +0.192$)

- Semantic errors: Minimal change ($\Delta\lambda_2 = -0.009$)

Models develop distinct computational pathways with unique geometric signatures for different tasks.

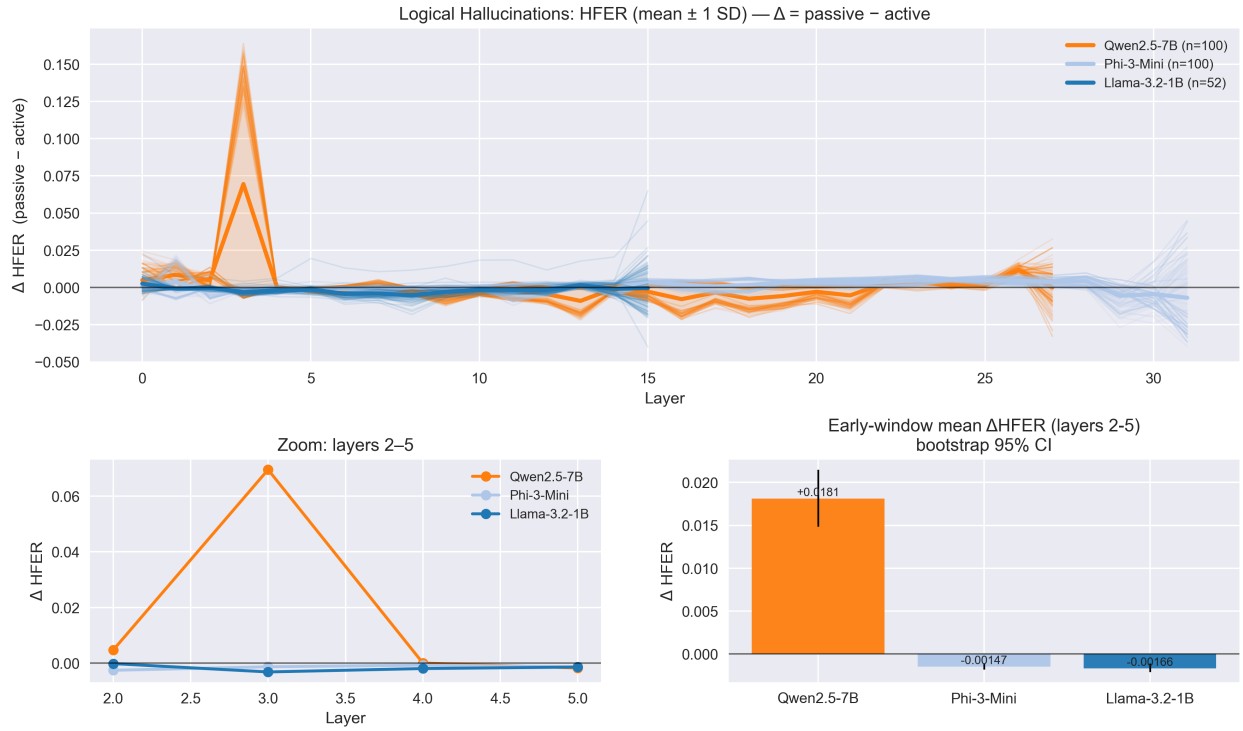

Figure 12: High-frequency energy ratio (Logical Reasoning).

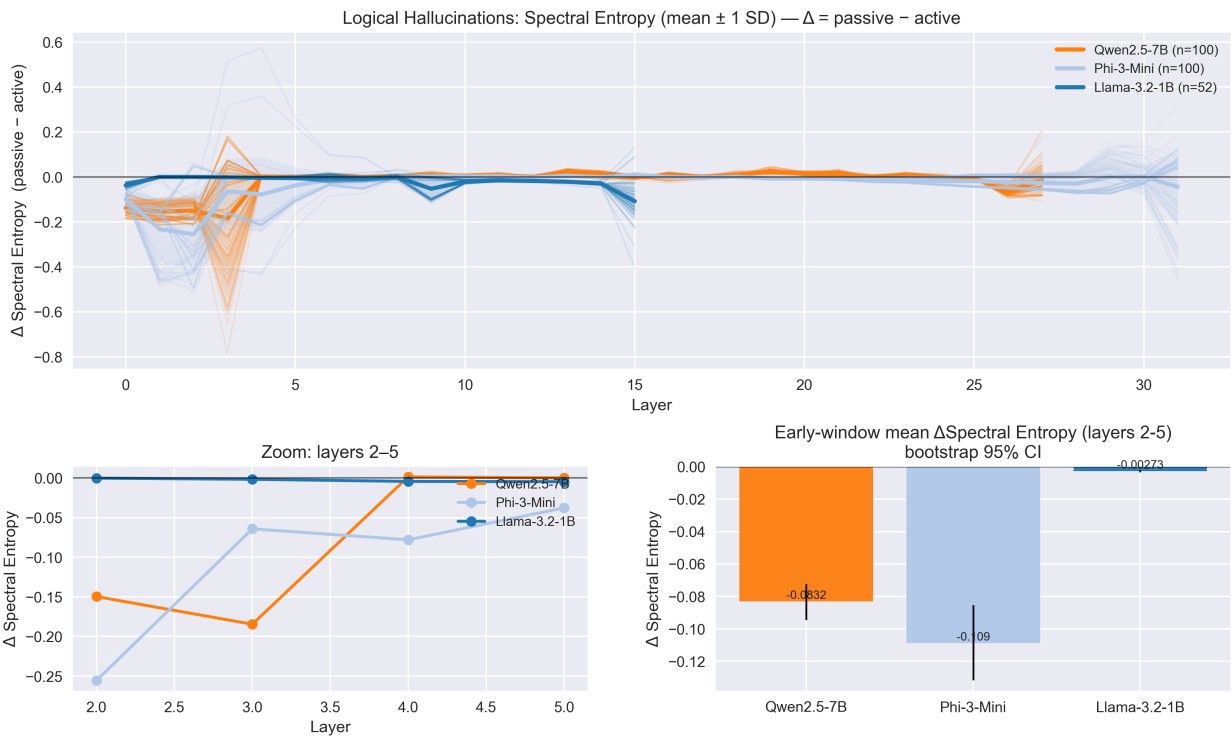

Figure 13: Spectral entropy (Logical Reasoning).

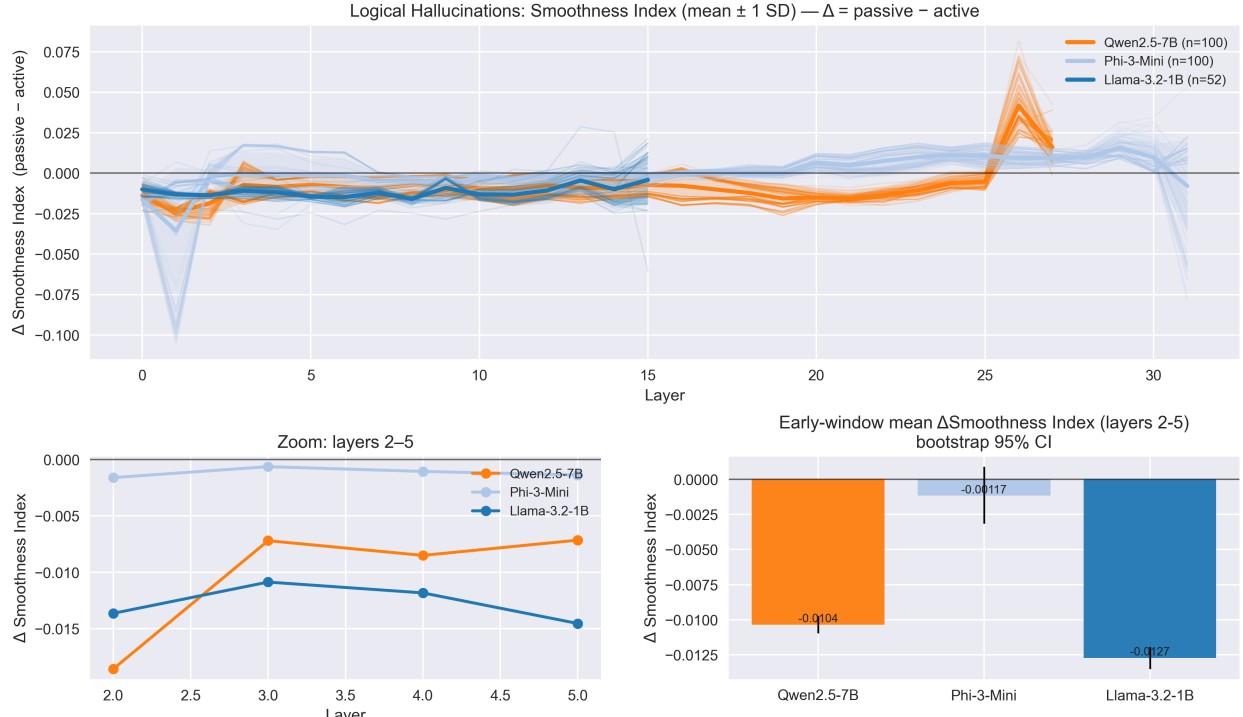

Figure 14: Smoothness index (Logical Reasoning).

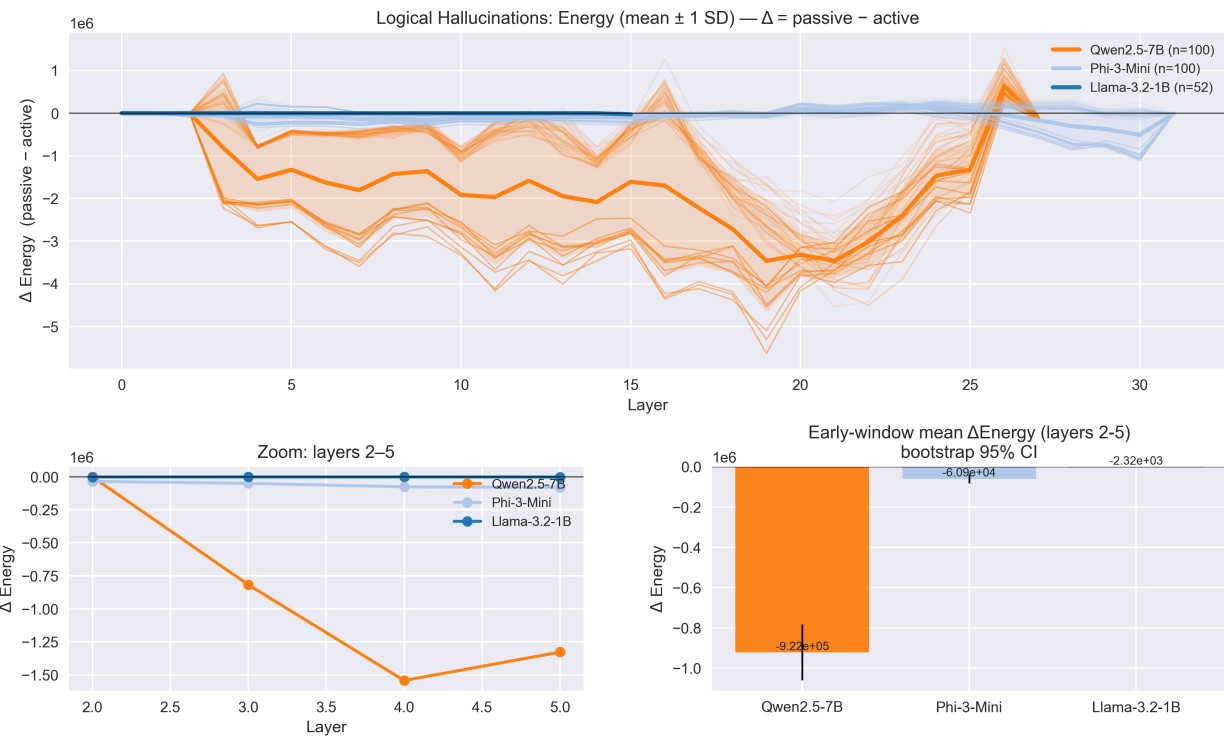

Figure 15: Compute proxy (Logical Reasoning).

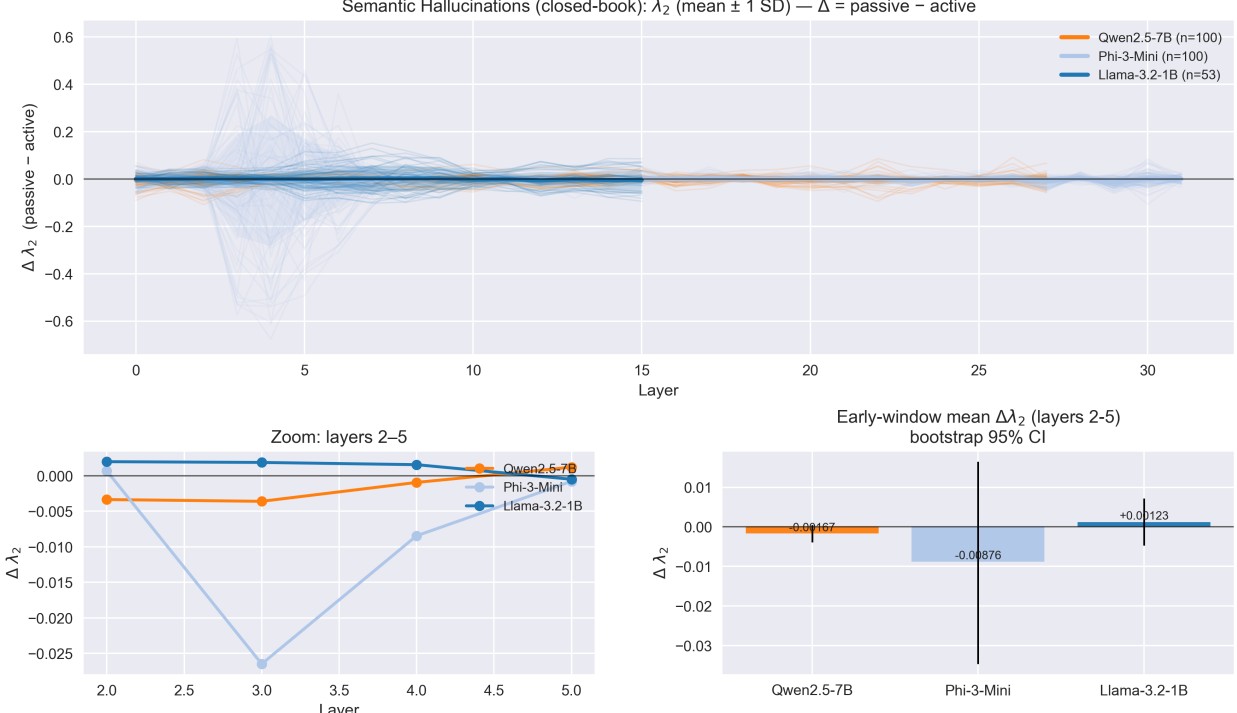

Figure 16: Early-window connectivity shift (Semantic Consistency). Layerwise $\Delta\lambda_2$ (inconsistent−consistent) per model.

## 5.2 Computational Economy of Errors

Across all architectures and domains, incorrect processing is computationally cheaper than maintaining consistency. Errors represent a "path of least resistance" that models follow under resource constraints.

## 5.3 Information Organization Spectrum

Different failure types show opposing geometric fingerprints:

- Syntactic failures: Connectivity collapse, increased entropy

- Logical failures: Hyperconnectivity, reduced entropy

- Semantic failures: Preserved connectivity, massive entropy increase

> **Takeaway**
>
> The framework generalizes beyond syntax to capture logic and semantics with domain-specific patterns. Same model + same metric + different phenomena = different signatures. This enables: (1) model fingerprinting, (2) vulnerability auditing, (3) failure mode diagnosis, (4) training data inference. A single-metric or single-domain approach would miss these distinctions.

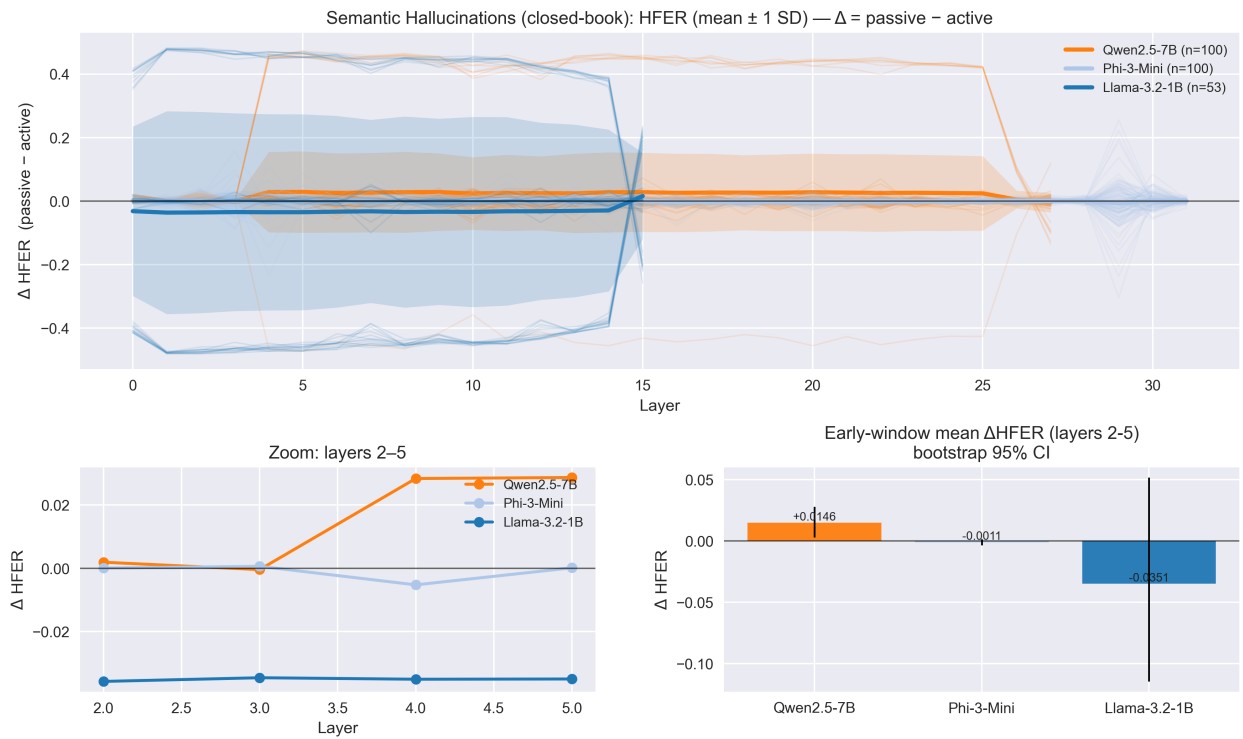

Figure 17: High-frequency energy ratio (Semantic Consistency).

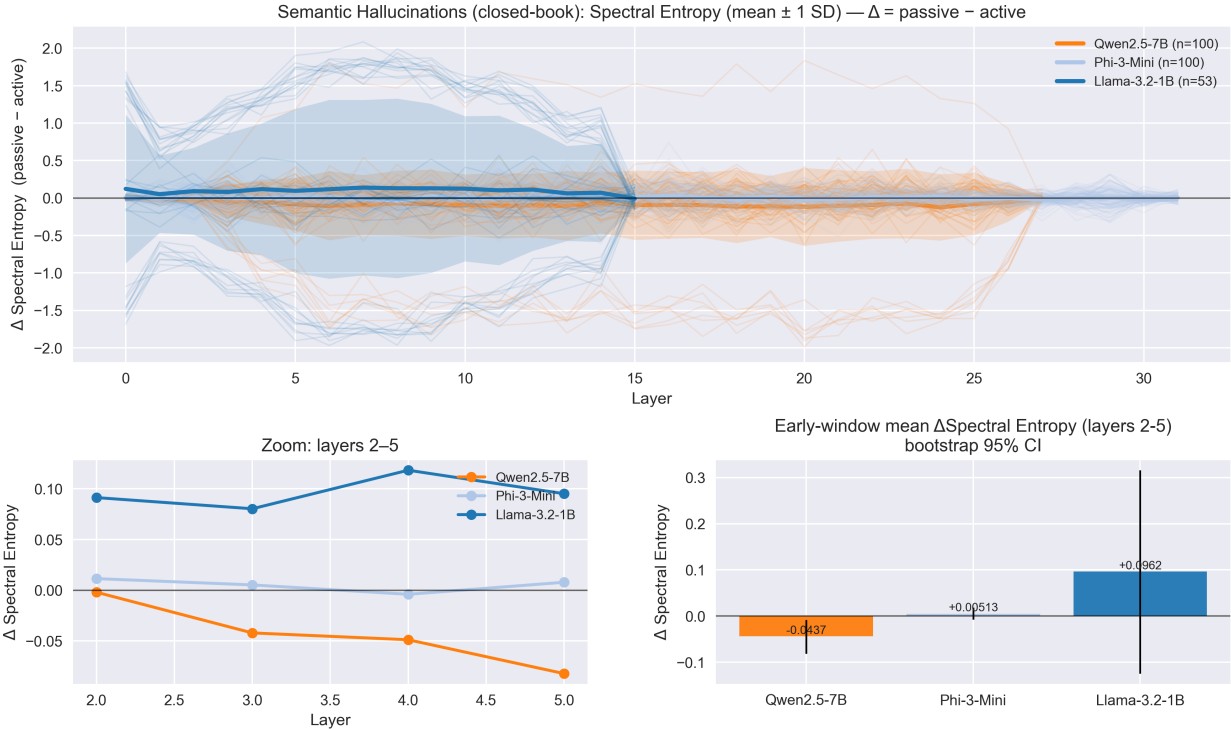

Figure 18: Spectral entropy (Semantic Consistency).

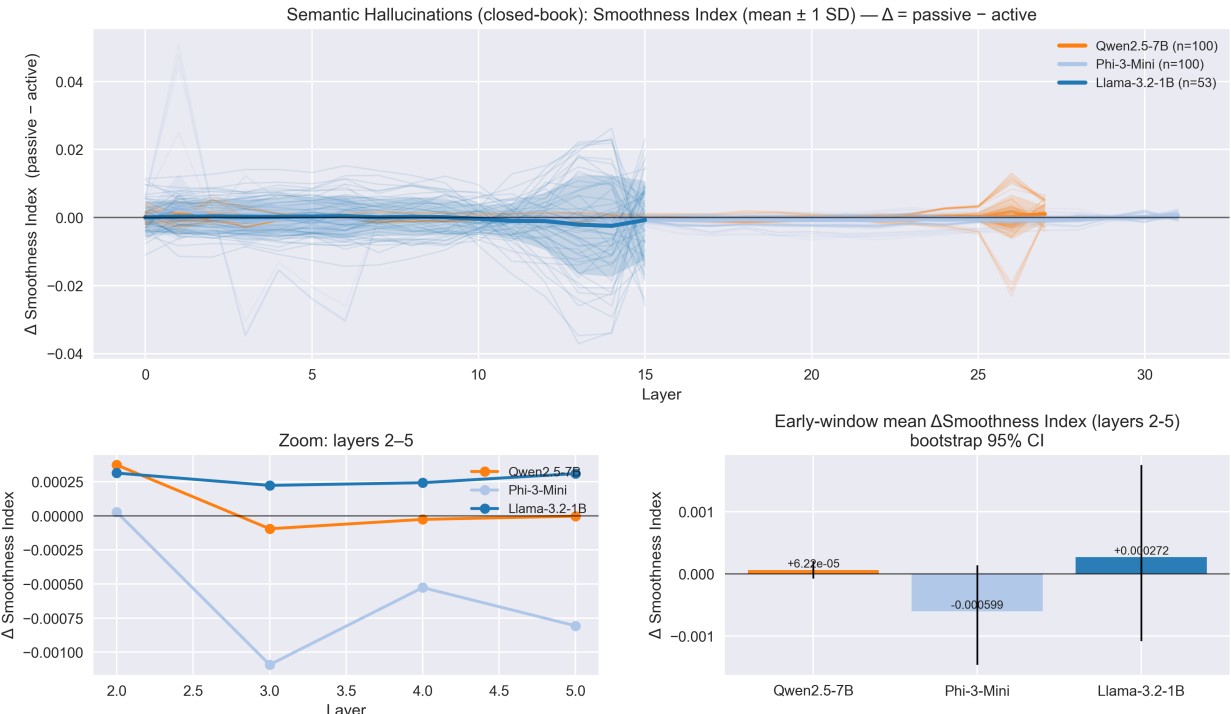

Figure 19: Smoothness index (Semantic Consistency).

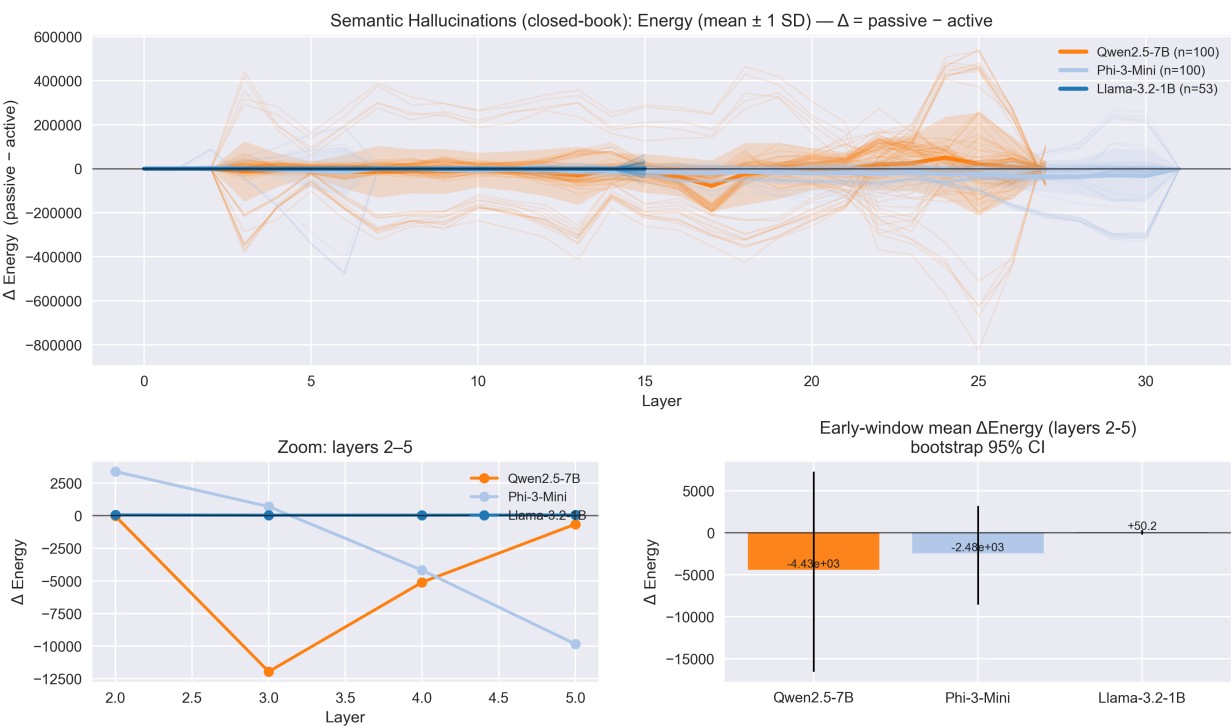

Figure 20: Compute proxy (Semantic Consistency).

# 6 Enlarged Main Paper Figures

Per Reviewer YXjU's request, we provide enlarged versions of main paper figures for improved legibility.

## 6.1 Phi-3-Mini Results

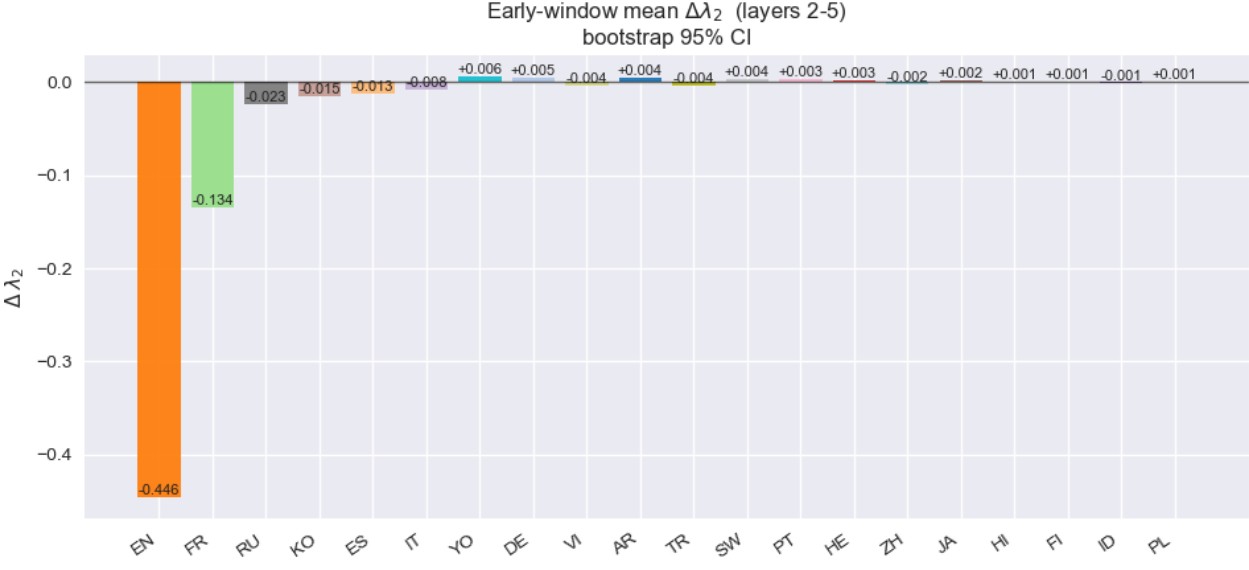

Figure 21: Phi-3-Mini: Per-language early-window $\overline{\Delta\lambda_2}_{[2,5]}$. English shows a large negative outlier ($\Delta\lambda_2 \approx -0.446$); all other languages cluster near zero.

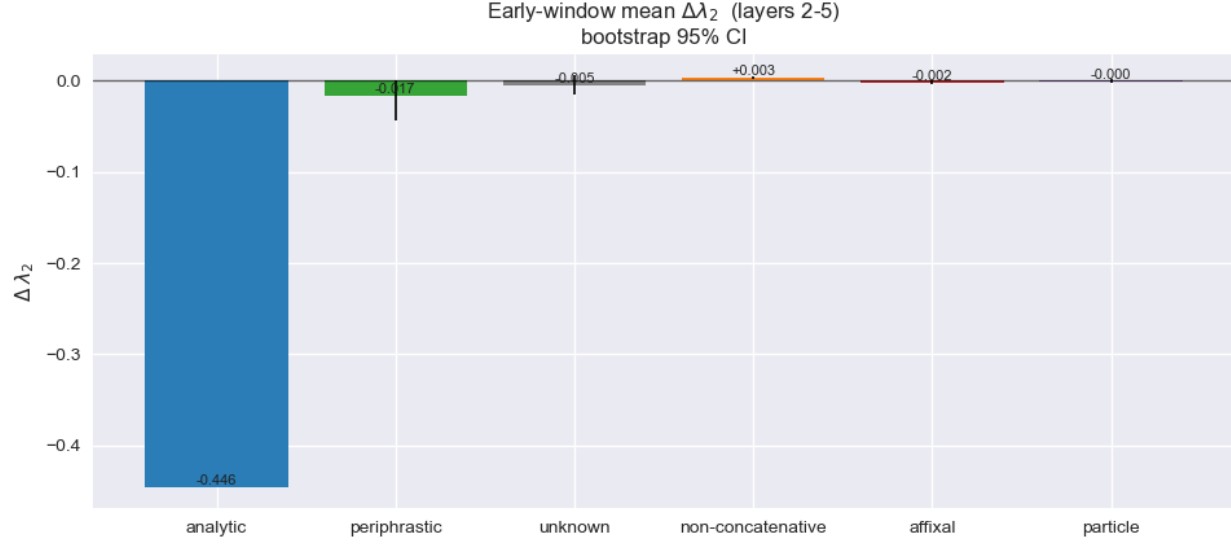

Figure 22: Phi-3-Mini: Early-window $\overline{\Delta\lambda_2}_{[2,5]}$ by voice type. The analytic type (English) shows the large negative effect; other types cluster near zero.

## 6.2 Qwen2.5-7B Results

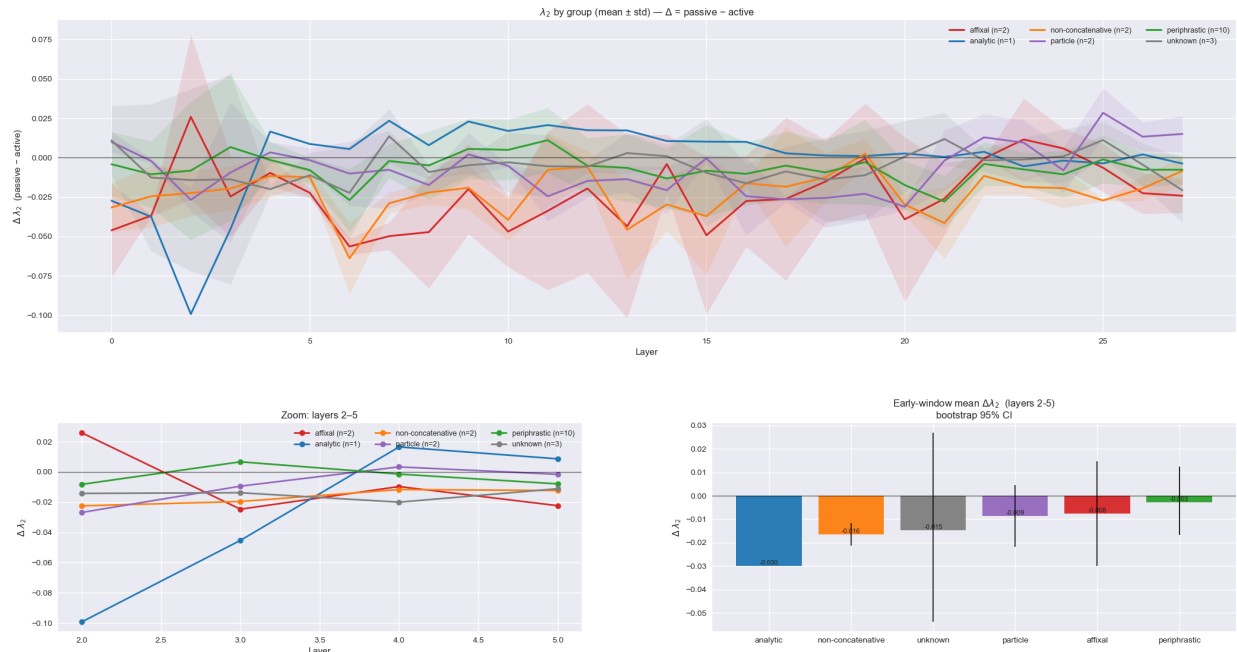

Figure 23: Qwen2.5-7B: Early-window $\overline{\Delta\lambda_2}_{[2,5]}$ by voice type. All types show small but consistent negative shifts, indicating robust multilingual processing.

## 6.3   LLaMA-3.2-1B Results

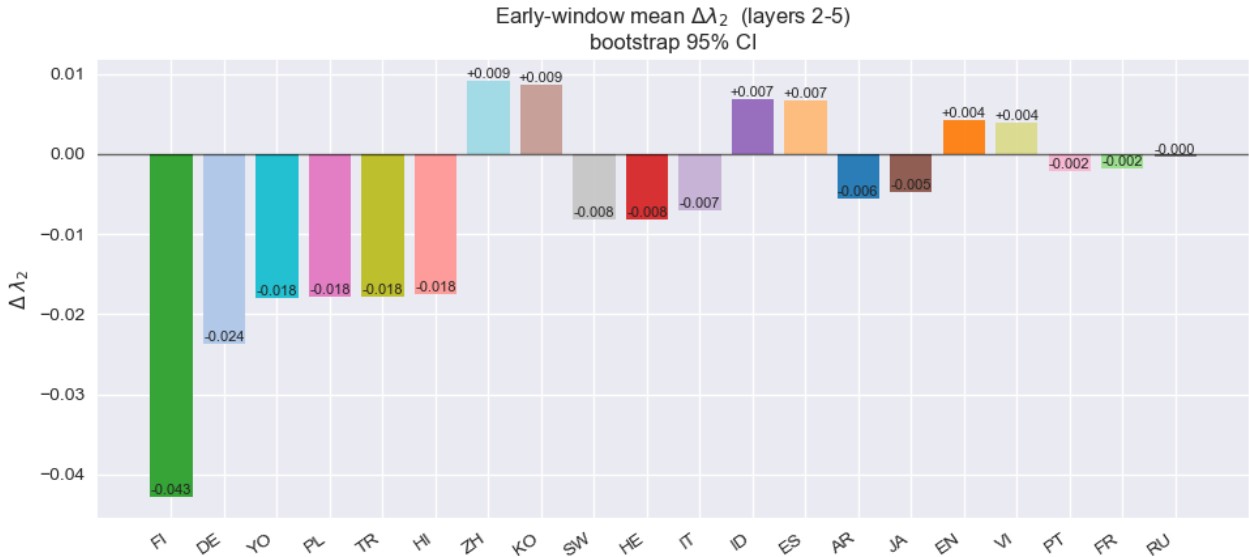

Figure 24: LLaMA-3.2-1B: Per-language early-window $\overline{\Delta\lambda_2}_{[2,5]}$. Mostly near zero with small negative values.

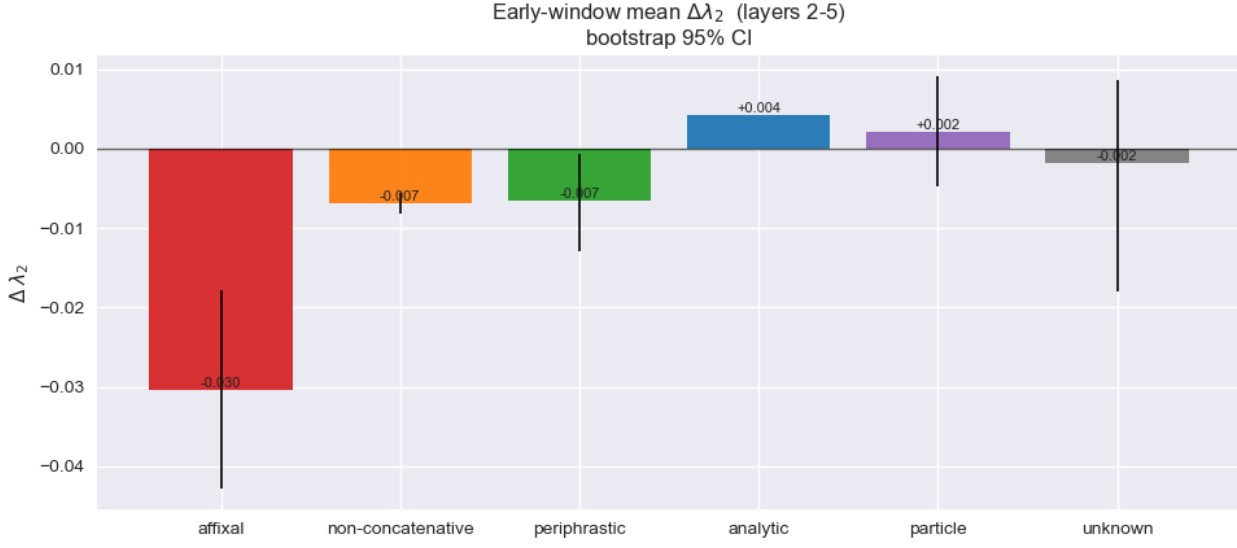

Figure 25: LLaMA-3.2-1B: Early-window $\overline{\Delta\lambda_2}_{[2,5]}$ by voice type. Small magnitudes but consistent ordering (affixal < periphrastic < analytic).

# 7 Summary: Addressing Reviewer Concerns

**R-YXjU: "Why layers 2-5?"** We demonstrate that layers 2-5 capture distinct signatures for voice, wh-questions, logic, and semantics. The same window reveals different patterns for different phenomena, proving it's a general diagnostic for early context integration.

**R-YXjU/R-oGBN: "Narrow scope / voice-specific"** We validate across four cognitive domains with construction-specific vulnerability patterns (opposite signatures for passive vs. wh-questions vs. logic).

**R-oGBN: "What can you do with this?"** Model fingerprinting, vulnerability auditing, failure mode diagnosis, training data inference, all without access to training data.

**R-2s1B: "Correlational, not causal"** Head ablation analysis (§1) provides causal evidence that early-layer heads directly maintain connectivity.

**All reviewers: "Weak/inconsistent results"** Apparent inconsistencies (Phi-3 collapse for passives but robustness for wh-questions) reflect genuine construction-specific vulnerabilities, a core finding, not a weakness. Effect sizes: Cohen's $d = 2.83$ for Phi-3 passive, $d = 1.12$ for logical hyperconnectivity.