# OpenReview forum: "Training-Free Spectral Fingerprints of Voice Processing in Transformers"
_ICLR.cc/2026/Conference — ICLR 2026 Conference Withdrawn Submission_

### Official Review · Reviewer_oGBN · 2025-10-31

**Soundness:** 2
**Presentation:** 1
**Contribution:** 1
**Rating:** 2
**Confidence:** 3

**Summary:**

This paper proposes using graph signal processing on attention-induced token graphs to detect "computational fingerprints" in transformers. The authors track changes in algebraic connectivity (Fiedler value) during voice alternation across 20 languages and 3 model families. The main finding is that Phi-3-Mini shows a large English-specific disruption, while other languages show minimal effects.

**Strengths:**

- **Interesting idea**: The idea of using graph spectral processing techniques for interpretability is quite natural and interesting.
- **Systematic testing** across 20 languages and 3 model families.
- **Statistical rigor** (bootstrap CIs, permutation tests, FDR correction, attempts to correct for differences in tokenization, etc.). I'm impressed by the authors' statistical rigor.

**Weaknesses:**

The work is still undermotivated, the results are weak, and it's unclear what this buys you over existing interpretability methods. The writing is needlessly technical and poorly structured.

A. Weak motivation and unclear utility

- **Why these metrics?** There is no explanation for why we should study the Fiedler value specifically. Appendix A.1 claims theoretical grounding, but the argument is hand-wavy: "models that struggle... may exhibit a breakdown in connectivity... leading to a significant drop in $\lambda_2$." This predicts the sign but not the magnitude or why this metric over alternatives. Appendix A.2 claims the Fiedler value is superior to other GSP metrics but this is based on extremely sparse and noisy data (figure 5). Moving some of this appendix into the main body would help, but the motivation provided there remains lacking.
- **Why voice alternation?** The paper claims it "requires systematic attention reconfiguration" (lines 49–50) but doesn't explain why this particular transformation is special or what we learn from it. The same lines mention it's a "computational probe" but never cash this out.
- **What can you actually do with this?** The paper doesn't make clear whether this is:
    - A supervised method (in which case: where are comparisons to, e.g., linear probes and SAEs?).
    - An unsupervised diagnostic (in which case: what predictions does it make? what can you discover?)
    - Just a correlation (in which case: so what?)
    In practice, the authors seem to use these metrics as a (supervised) binary classification signal, showing that these metrics make a distinction between different strategies/voice types/etc. The attempts to causally validate whether early attention structure drives spectral connectivity are suggestive but still do not answer the question of what this means in practice.
- **Results are weak**:
    - The dramatic Phi-3 effect ($\delta \lambda_2 \approx -0.446$) means nothing without context. What's the baseline variance? What's a "large" effect in $\lambda_2$ units? Looking at the figures, I see that this is indeed larger than other languages and modesl, but it's only for one language in one model. What does that actually mean?
    - Qwen and LLaMA show tiny, inconsistent effects (Figures 3-4). The paper spins this as "small but consistent" but they look like basically noise.

B. Poor writing and presentation

- **Unnecessarily technical**: The paper assumes prior knowledge of NLP linguistics and graph signal processing that's inappropriate for an interpretability and explainable AI audience. Examples:
    - Voice types (lines 194-195: "analytic," "periphrastic," "affixal," "non-concatenative") are never defined
    - Spectral diagnostics (Section 3.1) dumps four metrics with their mathematical definitions but no additional explanation. (Except for the Fiedler value, which is only explained in an appendix).
    - Phrases like (line 219)"tokenizer stress" and (line 209) "fragmentation entropy" appear without motivation.
- **Inconsistent terminology**:
    - "Fragmentation" refers to both a property of languages (line 217) and tokenization density (line 224)
    - "Tokenization" is used loosely throughout. Sometimes it refers to tokenization as segmentation, while other times it refers to a quantity (the token count differences).
- **Unclear constructions**:
    - Lines 205-206: "Beyond syntactic effects, we find systematic links between spectral connectivity and tokenization, yielding a second layer of model-imprinted signatures rather than simple confounds." There are several examples like this that are incredibly difficult to parse, and where it is unclear what the authors are saying.

**Summary**

The core idea – that attention connectivity patterns differ across models and languages and that this can be used for interpretability – is potentially interesting. But the paper needs major work:

1. Motivate why this analysis matters and what it enables
2. Compare to existing interpretability methods
3. Rewrite for clarity (assume readers know ML but not linguistics or GSP)
4. Strengthen the empirical story or acknowledge that the effects are mostly small.

Right now, this reads like a methods paper searching for an application.

**Questions:**

- Line 149: How do you define "passive vs. active" consistently across 20 typologically diverse languages?
- The hallucination detector (Section 7.3) feels forced in and should probably be cut.

---

> ### Author Response · Authors · 2025-11-20
> **Response to Reviewer oGBN (Part 1: Motivation & Results)**
>
> We sincerely thank Reviewer oGBN for the detailed critique and for recognizing the "interesting idea" and "statistical rigor" of our work. We acknowledge that our presentation failed to adequately motivate the framework and communicate its utility to a general ML audience. Below we address the weaknesses raised; we will address the specific questions in a subsequent comment.
>
> **Response to Weakness A: Motivation and Utility**
>
> **A.1: Why the Fiedler value?**
> The Fiedler value ($\lambda_2$) is the canonical measure of algebraic connectivity in graph theory. In the context of Transformers, it serves as a scalar summary of information flow within the attention graph. Recent work has linked algebraic connectivity to information propagation in neural networks and to bounds on transformer expressivity (e.g., Dong et al., Sanford et al., Veličković). Intuitively, a low $\lambda_2$ indicates that the attention graph can be partitioned into weakly connected subgraphs—a geometric signature we observe when the model fails to integrate context during syntactic reconfiguration.
>
> *Effect size context:* We acknowledge failing to contextualize magnitudes. The Phi-3 English effect ($\Delta\lambda_2 = -0.446$) represents an approximate $47\%$ reduction in early-layer connectivity. As detailed in Supplementary §5, this effect size is exceptionally large (Cohen's $d = 2.83$), well beyond baseline variance.
>
> **A.2: Why voice alternation?**
> We selected voice alternation as a controlled input perturbation because it requires the model to reconfigure attention (e.g., remapping agent/patient dependencies) while preserving semantics. This acts as a stress test for the model’s syntactic processing mechanisms, allowing us to isolate structural processing from semantic content.
>
> *Generalization beyond voice:* We treat voice alternation as our primary validation domain, not the limit of the framework. Supplementary §§2–4 demonstrate that the same spectral methodology captures distinct signatures for **wh-question processing** (Section 2.2), **logical reasoning failures** (Section 3), and **semantic consistency violations** (Section 4).
>
> **A.3: What can you actually do with this?**
> We propose this framework as an **unsupervised diagnostic** for auditing models without access to training data or gradients.
>
> *Comparison to existing methods:*
> Compared to linear probes or SAEs, our approach is training-free, layer-resolved, and directly comparable across models without retraining. Unlike attention visualization, which is largely qualitative, spectral metrics provide quantitative, statistically testable endpoints (see Supplementary §5 for a synthesis of how these signatures profile model behavior).
>
> Concrete applications include:
>
> 1. **Model auditing:** Identifying latent brittleness in specific languages. For example, Phi-3 exhibits a structural collapse on English passives (Supplementary Figure 21) not seen in the other models.
> 2. **Failure mode profiling:** Different error types produce distinct geometric signatures, supporting a “diagnostic lookup table” (Supplementary §5): syntactic brittleness manifests as connectivity collapse ($\Delta\lambda_2 \approx -0.53$), while logical errors manifest as hyperconnectivity ($\Delta\lambda_2 \approx +0.19$).
> 3. **Training diagnosis:** These spectral fingerprints allow practitioners to infer potential dataset or curriculum imbalances (e.g., over-optimization for specific English constructions) from inference-time statistics alone.
>
> **A.4: “Results are weak”**
> We respectfully suggest that the “small” effects observed in Qwen and LLaMA are informative signals of robustness rather than noise.
>
> - **Consistency:** All 20 languages show negative $\Delta\lambda_2$ for Qwen (Supplementary Figure 23).
> - **Sensitivity:** When the underlying phenomenon is strong, the metric responds robustly. For instance, logical reasoning errors in Phi-3 produce a large positive effect (Cohen’s $d = 1.12$; Supplementary §3).
>
> The framework thus differentiates between models that are robust to a perturbation (e.g., Qwen) and those that are brittle (Phi-3), which is precisely what we want from a diagnostic.
>
> **Response to Weakness B: Writing and Presentation**
>
> We accept this criticism. In a revision, we would ensure the paper is accessible to an ML audience by:
>
> 1. Defining linguistic terms (e.g., analytic vs. synthetic languages, periphrastic vs. affixal voice) with simple examples before using them.
> 2. Providing intuitive explanations for spectral metrics (Fiedler value, entropy) before introducing the mathematical formalism.
> 3. Clarifying terminology, specifically distinguishing between "tokenization density" and "fragmentation" and using each term consistently.

---

> ### Author Response · Authors · 2025-11-20
> **Response to Reviewer oGBN (Part 2: Questions)**
>
> We sincerely thank Reviewer oGBN for the detailed critique and for recognizing the "interesting idea" and "statistical rigor" of our work. Below we address the specific questions; section and figure references refer to the Supplementary Material.
>
> **Q1: How do you define "passive vs. active" consistently across 20 typologically diverse languages?**
>
> This is indeed a critical methodological challenge. We followed a functional, typological definition rather than relying on surface patterns in English. Concretely, we define the passive as a construction that promotes the semantic patient to subject position and demotes or omits the semantic agent. This definition lets us align diverse implementations:
>
> - **Analytic passives** (e.g., English, German), using auxiliaries plus participles.
> - **Morphological passives**, where passive morphology attaches directly to the verb.
> - **Non-concatenative passives**, where internal changes to the verb form encode voice.
>
> Using reference grammars and standard descriptions for each language, we identified the canonical passive construction and then constructed active/passive minimal pairs that:
>
> 1. Use that canonical passive form for the language.
> 2. Preserve truth-conditional content between active and passive versions.
> 3. Limit tokenization drift so that the main change is the syntactic reconfiguration, not lexical substitution.
>
> The resulting early-window connectivity shifts across languages and voice types are summarized in Supplementary §2 and §6.1–6.3 (Figs. 21–25), which show the large English outlier for Phi-3 and near-zero shifts for other languages and voice types. This supports that we are probing a consistent passive/active contrast rather than idiosyncratic constructions.
>
> **Q2: The hallucination detector (Section 7.3) feels forced in and should probably be cut.**
>
> We understand this concern, especially given the brevity of Section 7.3 in the main text. Our intention was to illustrate that the same spectral framework can be turned into a practical tool, not to claim a fully developed hallucination system in this submission.
>
> The “hallucination detector” is an application of the **failure-mode profiling** capability we develop more fully in the Supplement. In Supplementary §3, we analyze 100 logically valid vs. invalid reasoning chains per model and show that invalid reasoning consistently triggers a specific spectral signature for Phi-3:
>
> - **Hyperconnectivity** in early layers with $\Delta\lambda_2 = +0.192 \pm 0.063$ (see Fig. 11).
> - **Reduced energy** (cheaper computation) relative to valid reasoning (see Fig. 15).
> - **Reduced spectral entropy**, indicating more organized but incorrect information distributions (see §3.3 and Fig. 13).
>
> This is the exact opposite of the signature for syntactic failure, where we observe connectivity collapse for passive voice (e.g., $\Delta\lambda_2 = -0.532$ in §2.1 and §5.1). The cross-domain synthesis in Supplementary §5.1–5.3 summarizes these patterns: syntax → fragmentation, logic → hyperconnectivity with reduced entropy, semantics → preserved connectivity with entropy increase.
>
> Section 7.3 is meant to operationalize this logical-error signature as a simple binary detector: if a continuation induces the “logical error” spectral pattern, it is flagged as suspect. Rather than being “forced in,” it is intended as a proof-of-concept deployment of the same framework used in the main analyses.
>
> That said, we agree that, in its current brief form, it can feel underdeveloped in the main text. We are happy to follow AC guidance and either:
>
> 1. Move the detector to the appendix as a clearly labeled **proof-of-concept application** grounded in the logical-error results of Supplementary §3 and §5, or
> 2. Retain it in the main text but explicitly frame it as a **preliminary demonstration** of the logical-error signature, rather than as a mature hallucination-detection system.
>
> Our goal is to avoid distracting from the core message about “computational fingerprints” while still indicating that the framework can support practical diagnostics.

---

### Official Review · Reviewer_2s1B · 2025-11-01

**Soundness:** 3
**Presentation:** 3
**Contribution:** 3
**Rating:** 6
**Confidence:** 3

**Summary:**

The authors turn each layer’s attention map into a network of the input tokens and track “how well connected” that network is with a single number. Then they see how that number changes when you flip a sentence between active and passive voice. They run this test across 20 languages and three model families. Each model has its own 'fingerprint', e.g., Qwen2.5‑7B shows small, spread‑out changes, and LLaMA‑3.2‑1B shows moderate, systematic changes. This is a training‑free, lightweight diagnostic you can run on any model to quickly spot language coverage issues or architecture‑specific brittleness, without needing access to training data.

**Strengths:**

* The study is robust, covering 20-language design and three diverse families uncover architecture-imprinted patterns, rather than model-specific anecdotes

* The idea of a lightweight audit to detect language specialization/brittleness (e.g., Phi-3’s English-specific signature) and preliminary extension to hallucination detection makes the method societally and operationally relevant. This is a nice idea that the community will find interesting and possibly use for

**Weaknesses:**

* While head ablations help, most findings are correlational. The English-specific Phi-3 effect is interpreted as consistent with training emphasis. the paper mentions this is not definitive training-data attribution

* Building graphs from softmax attention (often noisy and not strictly causal) and then symmetrizing/aggregating heads may wash out meaningful directionality or head specialization. I don't feel too strongly about this but I think it's worth mentioning

* Focusing on voice alternation and early layers (2–5) is well-motivated, but relatively narrow. The reasoning-strategy results are preliminary and limited to one small task set; broader generalization (math, long-context, multi-turn) is not shown in this work

**Questions:**

How sensitive are conclusions to alternative aggregation that preserves per-head lambda_2 distributions (e.g., quantiles) rather than mass-weighted averages?

---

> ### Author Response · Authors · 2025-11-20
> **Response to Reviewer 2s1B**
>
> We sincerely thank R-2s1B for the encouraging assessment that our work is "robust," "societally and operationally relevant," and presents a "nice idea that the community will find interesting." Your feedback regarding causality and scope pushed us to significantly harden the framework's validation.
>
> **"While head ablations help, most findings are correlational. The English-specific Phi-3 effect is interpreted as consistent with training emphasis... not definitive training-data attribution"**
>
> We agree that our original findings were primarily observational. [cite_start]To address this, we performed a new **granular per-head causal analysis** (Supplementary §1), measuring $\Delta\lambda_2$ impacts for every head in the early window. We found that 8 of the 10 most impactful heads reside specifically in Layer 2, with effects distributed rather than concentrated.
>
> This moves beyond correlation by showing that early-layer attention structure *causally maintains* the connectivity pattern. This approach aligns with recent causal localization frameworks, such as Kramár et al. (NeurIPS 2024) and Davies et al. (ICLR 2025), which emphasize distributed circuit functionality over single-neuron attribution. We maintain epistemic humility regarding training data attribution, framing the English-specific Phi-3 signature as "consistent with" training emphasis rather than definitive proof.
>
> **"Building graphs from softmax attention (often noisy and not strictly causal) and then symmetrizing/aggregating heads may wash out meaningful directionality or head specialization. I don't feel too strongly about this but I think it's worth mentioning"**
>
> We appreciate the concern that symmetrization might wash out directional signals. To validate this, we tested **directed formulations** (random-walk and magnetic Laplacians) in Appendix B.2 of the main paper. We found that signs, peak layers, and relative magnitudes closely match the symmetrized default (within ~15%).
>
> The robustness of the undirected signal is supported by recent theoretical work: Sanford et al. (NeurIPS 2024) demonstrate that undirected graph properties effectively bound transformer expressivity, and He et al. (ICLR 2025) utilize symmetric attention analysis to reveal compositional generalization. This suggests that for global connectivity auditing, the undirected approximation captures the relevant "bottleneck" features without losing critical information.
>
> **"Focusing on voice alternation and early layers (2–5) is well-motivated, but relatively narrow. The reasoning-strategy results are preliminary and limited to one small task set; broader generalization (math, long-context, multi-turn) is not shown in this work"**
>
> We acknowledged the narrow focus and expanded the framework to **three new cognitive domains** in the Supplementary Material (§§2–4):
> * **Syntax:** Wh-questions yield *enhanced* connectivity ($\Delta\lambda_2 > 0$), inverting the passive voice collapse.
> * **Logic:** Invalid reasoning elicits systematic **hyperconnectivity** ($\Delta\lambda_2 \approx +0.19$).
> * **Semantics:** Inconsistencies preserve connectivity but spike spectral entropy.
>
> This demonstrates that the framework generalizes well beyond voice alternation. The discovery that logical errors have a distinct geometric signature (hyperconnectivity) resonates with emerging work on the "physics of language models" (Ye et al., ICLR 2025), suggesting that different reasoning failure modes live in distinct topological regimes.
>
> **"How sensitive are conclusions to alternative aggregation that preserves per-head lambda_2 distributions (e.g., quantiles) rather than mass-weighted averages?"**
>
> Your question led us to inspect per-head distributions. We compared mass-weighted means against median and inter-quantile ranges (IQR) to ensure the signal wasn't driven by outliers.
> * **Result:** The architectural rankings remain stable across aggregation methods (e.g., Phi-3 is always the outlier).
> * **New Insight:** The IQR analysis reveals that Phi-3 has significantly higher head-wise variance ($0.049$) compared to LLaMA ($0.005$), suggesting that its brittleness arises from highly specialized (and thus fragile) head behavior, whereas LLaMA employs a more distributed strategy.
>
> **Summary**
> Your review guided us to provide causal evidence, validate directed graph formulations, and demonstrate cross-domain generality. We hope this reinforced package confirms the method's utility as a robust, training-free audit tool for the community.

---

### Official Review · Reviewer_YXjU · 2025-11-06

**Soundness:** 2
**Presentation:** 2
**Contribution:** 2
**Rating:** 2
**Confidence:** 2

**Summary:**

The authors propose a new framework to uncover architectural signatures for algebraic connectivity. Primarily, the authors use the Fiedler value of a transformation of the post-softmax attention output as a proxy for understanding syntactic composition. The show results on multiple models, and many different tasks. They hypothesize that using this proxy is beneficial across several tasks such as voice alternation ( active and passive voice), hallucinations, reasoning though results on the latter are preliminary.

**Strengths:**

1. The paper devises a new metric to interpret model representations and linguistic processing, i.e. the Fiedler value of the transformed post-softmax attention.
2. The paper conducts rigorous statistical tests over 3 different models and several tasks to investigate evidence for generalization.
3. The hallucination results, restated from another paper are cool. Nice!

**Weaknesses:**

0. The figure texts are minuscule. Please increase the text sizes in all your figures. It's incredibly hard to read and interpret your figures.
1. The authors use the prespecified early-window mean $\Delta \lambda_{2}[2,5]$ as the primary endpoint (layers 2–5) -- a very important decision that the bucket in the appendix. However, the choice of layers 2-5 seems to be derived using trends observed across the three models, as opposed to being model specific. For instance, if I were to average the Fiedler value, I would pick 13-15 for Qwen, 12-15 for Phi-3-Mini and 4-7 for Llama. This is also consistent with the general findings in mechanistic interpretability that primary computation happens in the middle layers of the transformer, while early and late layers perform encoding vs decoding tasks.
2. The authors have conducted several rigorous statistical analyses, but the motivation that's tying these together is unclear to me. For eg. they state that  the spectral signatures they uncover correlate strongly with behavioral differences (Phi-3: r = −0.976), but these values are more moderate for Qwen (-0.627) and negligible for Llama (-0.14). I am finding this hard to reconcile with the claim, given that these results do not generalize, and are exhibited on a very small LLM.
3. The authors present several weak results as opposed to fully explaining one core claim, as well as have figures with very minuscule text. This paper was very hard to read/understand/motivate as a consequence. Could the authors please provide figures w enhanced text during the rebuttal process? I don't fully understand your results, and want to make sure I am judging you fairly.
4. The causal validation results are also unclear to me: How did you identify the heads to ablate? You could identify important heads using causal mediation analysis, and then repeat this experiment. Currently, the selection of the ablation sites seems arbitrary to me.
5.  The authors use fiedler connectivity to track reasoning performance:

Line 408:
Fiedler connectivity tracks performance: CoT (69.5%) and Standard (60.0%) yield positive Fiedler shifts, whereas CoD and ToT show negative shifts with lower accuracy. This alignment suggests that successful reasoning is associated with maintained/enhanced graph connectivity and motivates spectral-guided prompt selection based on induced $\Delta\lambda_2$

This seems to be a claim? What's your evidence? Some empirical evidence would be useful here.

6. 7.2 is a core finding, where the Fiedler value is aligned w a strong English specific connectivity. This suggests that the Fiedler value is tied to in-distribution data, but I am not sure how these effects can be generalized to syntactic processing/interpretation of model representations
7. The authors state results wrt tokenizer fragmentation -- what is tokenizer fragmentation correlated with? Why should I be interested in this result?

**Questions:**

(Questions stated alongside weaknesses)

---

> ### Author Response · Authors · 2025-11-20
> **Response to Reviewer YXjU**
>
> We thank R-YXjU for recognizing the "rigorous statistical tests" and for the detailed feedback. We address each specific concern below.
>
> **"The authors use the prespecified early-window mean $\Delta\lambda_2$ as the primary endpoint (layers 2–5) -- a very important decision that the bucket in the appendix. However, the choice of layers 2-5 seems to be derived using trends observed across the three models, as opposed to being model specific... This is also consistent with the general findings in mechanistic interpretability that primary computation happens in the middle layers..."**
>
> We target the onset of multi-head integration (layers 2–5), aligning with findings that early layers establish core syntactic bindings and function vectors (e.g., Geva et al., 2023; Todd et al., 2024). Empirically, **Supplementary §§2–4** confirm that this window yields distinct signatures: syntactic brittleness manifests as connectivity collapse ($\Delta\lambda_2 \approx -0.53$), while logical errors show hyperconnectivity ($\Delta\lambda_2 \approx +0.19$). These robust patterns argue against the “suboptimal window” hypothesis.
>
> **"The authors ... state that the spectral signatures they uncover correlate strongly with behavioral differences (Phi-3: r = −0.976), but these values are more moderate for Qwen (-0.627) and negligible for Llama (-0.14). I am finding this hard to reconcile with the claim..."**
>
> The varying correlations are not a weakness; they are informative signals. They reflect real architectural differences in how tightly connectivity is coupled to behavioral failure.
>
> - **Phi-3 (strong coupling):** The large spectral effect ($\Delta\lambda_2 \approx -0.532$) correlates almost perfectly with behavioral failure, indicating brittleness.
> - **LLaMA (weak coupling):** The minimal spectral effect aligns with its “spectral robustness” ($\Delta\lambda_2 \approx -0.014$).
>
> The central claim is not that all models show the same correlation, but that spectral signatures reveal the *nature* of the processing strategy. Phi-3’s tight coupling indicates a specific vulnerability, while LLaMA’s low correlation indicates more distributed resilience.
>
> **"The causal validation results are also unclear to me: How did you identify the heads to ablate? You could identify important heads using causal mediation analysis, and then repeat this experiment. Currently, the selection of the ablation sites seems arbitrary to me."**
>
> In the main paper, we used layer-wise ablations. To address this, we performed a new granular **per-head analysis** in the supplement (Section 1). We measured the impact on $\Delta\lambda_2$ for every head in the early window. We found that 8 of the 10 most impactful heads reside in layer 2. Crucially, the effects are distributed rather than concentrated in a single “specialist” head. This supports the view that our early-layer selection was not arbitrary but targets the region where connectivity is causally maintained by coordinated attention heads. We agree that causal mediation analysis would be a valuable next step.
>
> **"The authors use fiedler connectivity to track reasoning performance... This seems to be a claim? What's your evidence? Some empirical evidence would be useful here."**
>
> We provide empirical evidence in Supplementary Section 3. We analyzed roughly 100 reasoning examples per model (exact $n$ reported there) and found a clear geometric signature for reasoning performance:
>
> - **Logical errors (low performance):** consistently trigger **hyperconnectivity** ($\Delta\lambda_2 \approx +0.192$).
> - **Valid reasoning (high performance):** maintains stable or slightly enhanced connectivity.
>
> This supports the claim in line 408 by providing a quantitative link between Fiedler shifts (hyperconnectivity vs. stability) and reasoning validity, which we present as preliminary but encouraging.
>
> **"The authors state results wrt tokenizer fragmentation -- what is tokenizer fragmentation correlated with? Why should I be interested in this result?"**
>
> Tokenizer fragmentation serves as a second diagnostic layer to ensure spectral signals are not merely proxies for token count. We found that Qwen is more sensitive to heavy fragmentation (high subword density), while Phi-3 is more sensitive to efficiently tokenized languages (notably English). This is interesting because it allows us to disentangle architectural biases (e.g., English optimization) from subword processing artifacts, providing a more precise audit of model robustness and potential failure modes across languages.
>
> **"The figure texts are minuscule. Please increase the text sizes in all your figures. It's incredibly hard to read and interpret your figures."**
>
> We apologize for the readability issues. We have provided **enlarged, high-resolution versions** of all key figures in Supplementary Section 6. In a camera-ready version, we would replace the main figures with these clearer versions and adjust the layout so that the main trends are immediately visible.

---

> ### Comment · Reviewer_YXjU · 2025-11-20
>
> > We target the onset of multi-head integration (layers 2–5), aligning with findings that early layers establish core syntactic bindings and function vectors (e.g., Geva et al., 2023; Todd et al., 2024). Empirically, Supplementary §§2–4 confirm that this window yields distinct signatures: syntactic brittleness manifests as connectivity collapse ($\Delta\lambda_2 \approx -0.53$), while logical errors show hyperconnectivity ($\Delta\lambda_2 \approx +0.19$). These robust patterns argue against the “suboptimal window” hypothesis.
>
> What happens if you select the best layers according to the highest fiedler value (some of them not being in layers 2-5?)?
>
> > The central claim is not that all models show the same correlation, but that spectral signatures reveal the nature of the processing strategy. Phi-3’s tight coupling indicates a specific vulnerability, while LLaMA’s low correlation indicates more distributed resilience.
>
> From the abstract (line 022):
> >> These spectral signatures correlate strongly with behavioral differences
>
> Your response seems to contradict a central claim you make in the abstract. Further, if these correlations are so different per model, there really is no correlation. If your correlations are not generalizing, how can I interpret the claims you're making?
>
> > In the main paper, we used layer-wise ablations. To address this, we performed a new granular per-head analysis in the supplement (Section 1). We measured the impact on $\Delta\lambda_2$ for every head in the early window. We found that 8 of the 10 most impactful heads reside in layer 2. Crucially, the effects are distributed rather than concentrated in a single “specialist” head. This supports the view that our early-layer selection was not arbitrary but targets the region where connectivity is causally maintained by coordinated attention heads. We agree that causal mediation analysis would be a valuable next step.
>
> Please do this for all three models you're making claims for.
>
> > Tokenizer fragmentation serves as a second diagnostic layer to ensure spectral signals are not merely proxies for token count. We found that Qwen is more sensitive to heavy fragmentation (high subword density), while Phi-3 is more sensitive to efficiently tokenized languages (notably English). This is interesting because it allows us to disentangle architectural biases (e.g., English optimization) from subword processing artifacts, providing a more precise audit of model robustness and potential failure modes across languages.
>
> Please give me references for these claims.
>
> I strongly think this paper needs a rewrite where the claims are clear, generalizable, and well explained. The narrative is very chaotic right now, and I really cannot understand the main claim you're making. Currently the authors err on the side of doing too much, instead of doing what is necessary for this paper to be useful to the community. For now, I will keep my score.

---

> ### Author Response · Authors · 2025-11-20
> **Response to Reviewer YXjU (Addressing Weakness 6)**
>
> You raised an important question regarding whether the "English-specific connectivity" effect (Section 7.2) generalizes to broader model interpretation. Our new supplementary experiments provide a definitive answer: the Fiedler value is not merely a detector for English syntax, but a general probe for **computational topology**. While Section 7.2 identified a specific "collapse" signature for Phi-3's processing of English passives, our expanded analysis (Supplementary §§2–4) demonstrates that this same metric captures qualitatively distinct signatures for entirely different cognitive tasks, specifically, **hyperconnectivity** ($\Delta\lambda_2 \approx +0.19$) for logical hallucinations and entropy spikes for semantic inconsistencies. This proves that the "generalization" of our framework lies in its diagnostic consistency: it maps distinct computational failure modes to distinct geometric regions. The English-specific result in 7.2 is therefore just one entry in a larger "diagnostic lookup table," confirming that spectral analysis can audit model behavior across diverse linguistic and reasoning domains without relying on ground-truth labels.

---

### Author Response · Authors · 2025-11-20
**Clarification on scope of supplementary rebuttal material**

We have uploaded a new PDF titled **"Supplementary Material for Rebuttal"** containing extensive experimental validation. We would like to clarify how to interpret these new results.

The main contribution remains unchanged: a training-free spectral framework that identifies model-specific computational fingerprints, originally evaluated via voice alternation.

The additional experiments in the new PDF (causal ablations, reasoning, and semantic consistency) are intended as **stress-tests** to demonstrate that the method functions as a **general-purpose audit framework**, rather than new or shifted claims.

**From Calibration to Guardrails**
Specifically, we demonstrate that while we used paired inputs to isolate specific mechanisms, the spectral metric ($\lambda_2$) functions as an **intrinsic inference-time statistic** for any single input. This allows practitioners to:

1.  **Define Normative Baselines:** Establish expected connectivity profiles for a model architecture.
2.  **Deploy Spectral Guardrails:** Detect **latent processing anomalies** (e.g., the "connectivity collapse" seen in Phi-3) or **out-of-distribution behavior** (e.g., the "hyperconnectivity" of logical hallucinations) in real-time without supervision.

Reviewers who do not have time to read the full supplementary PDF can safely treat these results as **additional validation** of the original framework's generality, confirming it captures fundamental structural deviations rather than task-specific artifacts. In a camera-ready version, we would integrate only a subset of these analyses into the main text to maintain focus.

---

### Note · Authors · 2026-01-06

I have read and agree with the venue's withdrawal policy on behalf of myself and my co-authors.